# Coregistration of heading to visual cues in retrosplenial cortex

Kevin K. Sit[1] & Michael J. Goard [1,2,3] ✉

Spatial cognition depends on an accurate representation of orientation within an environment. Head direction cells in distributed brain regions receive a range of sensory inputs, but visual input is particularly important for aligning their responses to environmental landmarks. To investigate how population-level heading responses are aligned to visual input, we recorded from retrosplenial cortex (RSC) of head-fixed mice in a moving environment using two-photon calcium imaging. We show that RSC neurons are tuned to the animal's relative orientation in the environment, even in the absence of head movement. Next, we found that RSC receives functionally distinct projections from visual and thalamic areas and contains several functional classes of neurons. While some functional classes mirror RSC inputs, a newly discovered class coregisters visual and thalamic signals. Finally, decoding analyses reveal unique contributions to heading from each class. Our results suggest an RSC circuit for anchoring heading representations to environmental visual landmarks.

A sense of direction is essential for animals to accurately perceive and move through their environment. Rather than being coupled to absolute coordinates, such as magnetic north, direction is represented in the brain as the relative orientation of an animal in its current environment, termed "heading". Although heading and head direction are generally aligned in a freely moving animal, they can be experimentally decoupled when head movement and environmental movement are independent. For example, researchers have observed heading responses in insects that track the movement of a virtual environment even with the head immobilized[1-3].

A distributed network[4-8] of head direction (HD) neurons found throughout the mammalian brain represent direction through increased firing when an animal's head is in a specific orientation relative to the environment[4,9-12]. A central hub of this network is the anterodorsal thalamic nucleus (ADN), which contains a high percentage of HD neurons and is crucial for navigation[4,12-14]. Early investigations of responses in the HD cell network demonstrated that the vestibular system plays a critical role in both the generation and maintenance of HD neurons. Lesions of the vestibular system, either in sensory organs[15-17] or in upstream relay nuclei[18,19], degrade the

responses of HD neurons in ADN and other regions in the HD cell network, resulting in navigational deficits. Although HD cell responses have been observed in restrained rodents in virtual environments, all experiments to this point have involved physical head rotation, either animal-controlled[20-23], or experimenter-controlled[24-26].

Despite the importance of vestibular input, previous work has shown that the visual system, though not necessary for the generation of HD cell responses, plays a crucial role in anchoring the responses to cues in the external environment[4,10,24,27]. Experiments which remove visual information by recording from animals in darkness have shown that HD neuron firing gradually drifts from the animal's physical head direction over time[10,27]. Moreover, turning on the lights results in a nearly instantaneous realignment of HD cell responses to visual cues, even if the cues were inconspicuously moved in the darkness. This suggests that visual information exerts a dominant influence on representations, supplanting cues from other sensory systems[28]. Indeed, behavioral experiments in several species that either dampen or remove vestibular information have shown that animals are still capable of successful navigation through the use of visual landmarks alone[1,16,29-31]; though other sensory inputs, such as optic flow[21,32],

[1]Department of Psychological and Brain Sciences University of California, Santa Barbara, Santa Barbara, CA 93106, USA. [2]Department of Molecular, Cellular, and Developmental Biology University of California, Santa Barbara, Santa Barbara, CA 93106, USA. [3]Neuroscience Research Institute University of California Santa Barbara, Santa Barbara, CA 93106, USA. ✉e-mail: michael.goard@lifesci.ucsb.edu

proprioception[33], motor efference copy[34], and olfaction[35] also contribute. Although visual input is known to play a critical role in registering heading representations to the environment, the mechanism of alignment has not been fully elucidated. We define this mechanism as the process by which visual information is used to reference the animal's internal compass, so that the internal spatial maps properly represent the external physical world.

Recent research has suggested that the retrosplenial cortex (RSC), a cortical region in the HD cell network, may play an important role in the integration of these signals[6,7]. Lesioning RSC degrades HD cell responses in ADN[36,37], suggesting that RSC directly contributes to HD cell tuning, which is further supported by reciprocal projections from RSC to ADN[38–41]. Furthermore, RSC is one of only two principal areas, the other being the postsubiculum[38,39,42], that receives direct innervation from both the visual cortex and the ADN. RSC also has important roles in navigation, particularly regarding landmark detection[43–48] and path integration[49,50], and has been shown to integrate visual information with vestibular input in the angular head velocity (AHV) system[51]. Finally, in the presence of conflicting visual landmarks, individual HD neurons in RSC exhibit multimodal tuning curves that capture the angular offset with respect to each landmark, suggesting that RSC can represent and resolve this ambiguity[46,52]. These findings suggest that RSC is uniquely situated to receive and integrate sensory information from the visual system into a unified representation of heading.

Here, we use 2-photon calcium imaging of RSC cell bodies and axonal inputs to investigate how visual cues are used to register heading to the external environment.

## Results

### Neurons in the RSC represent heading in the absence of physical head movements

In order to investigate the interaction of visual input and heading using 2-photon imaging of somata and axon terminals, we developed a preparation in which mice are head restrained in a moving environment (Fig. 1A). To clarify our terminology, in previous studies the relative orientation of the head to the environment was generally determined by physical movement of the animal's head in a stationary environment, leading to the term "head direction cells". Since our experiments have the head immobilized, we define tuned neurons as "heading cells", since they represent the relative orientation of the stationary head to a moving environment. Heading responses have been found in invertebrates in preparations in which the head is immobilized[1–3], but to date all experiments investigating HD cells in mammals have incorporated physical head movement[20–23]. Virtual reality systems[22,53–55] are commonly used for navigation tasks in head-fixed preparations, but have some significant drawbacks, such as a lack of depth cues, reduced somatosensory feedback, and limited proprioceptive input. To address these limitations, we opted to use a walled circular chamber that floats on an air table[56] and contains an array of sensors to collect measurements from the chamber, including linear position, angular orientation, and speed (Fig. 1A, D, see "Methods"). This approach holds vestibular input constant, while allowing the mice to experience the visual and proprioceptive inputs associated with movement relative to the environment.

To determine if mammalian RSC neurons show tuning for heading even when the head is stationary, we allowed head-fixed mice to ambulate within a floating chamber while simultaneously imaging from the posterior dysgranular RSC ($n = 4$ recordings over 3 mice, Fig. 1B). Posterior dysgranular RSC was chosen because it receives direct projections from visual cortex[40,48,57]. To avoid excessive smoothing of heading responses due to the slow temporal dynamics of calcium imaging, we deconvolved the calcium traces and inferred spikes for all analyses (Fig. 1C). We then measured the response of each neuron as a function of the angular position of the cue card, revealing that a proportion of neurons in RSC faithfully represent heading (14.0%, 373/2665 neurons; Fig. 1D, E), similar to the proportion found in

freely moving rodents[6,7]. However, we observed that mice did not always fully sample the entirety of the cage in each recording (Figure S1). In addition, mice may have been able to use other sensory cues besides vision, such as odor, to determine their heading.

To isolate the visual contribution to heading representations, we transitioned to a controlled rotation design. We used guide bearings to limit translation while rotating the chamber with a motorized wheel (Fig. 1F). This approach has three distinct advantages: (1) it creates a regular trial structure in the data, (2) it ensures equal sampling of the entire range of headings, and (3) it reduces changes in the appearance of the visual cues due to variable proximity during chamber translation. To take advantage of the trial structure of our data, we created a new method to determine if a neuron was heading selective with a low false positive rate (Figure S2, see "Methods"). Using this method, we found a similar percentage heading neurons as in the experiments without controlled rotation, comprising 14.2% (452/3179 neurons) of the total number of cells recorded ($n = 16$ recordings over 5 mice, Fig. 1G, Figure S3). We wanted to ensure that the recorded cells were not simply activated by the passing of the visual cue over a visual receptive field. We aligned all the cells using their cross-validated preferred direction (see "Methods"), showing that heading-responsive cells tile the entire circumference of the chamber, even when the cue card is outside of the visual field of the mouse; though we note there is a moderate overrepresentation of responses when the cue is centered in front of the mouse near 0° (Fig. 1H). Together, these results suggest that the visual and proprioceptive input from the rotating chamber sufficiently drive dysgranular RSC neurons to accurately represent heading, even in the absence of vestibular modulation.

### Heading cells exhibit similar preferred directions during rotation of the animal or the environment

Previous research has emphasized the importance of vestibular information for the function of the heading network[58], so we next investigated whether the same neurons exhibit matched tuning during head fixation in a rotating environment versus during rotation of the head in a fixed environment. To address this, we added a belt-driven rotation collar to the previous experimental set-up, allowing physical rotation of the mouse head plate with the window centered underneath the imaging objective (Fig. 2A). The rotation had a constant, low angular velocity to allow the animal to walk along as the head plate was rotated. This approach allowed us to compare responses in the same neurons across two conditions: (1) in which the head rotates but the chamber is fixed or (2) in which the chamber rotates while the head is fixed ($n = 5$ recordings across 4 mice; Fig. 2B). Since the head is physically rotating under the microscope objective, we first registered each frame to a template calculated from the average frame during the chamber rotation recording (Fig. 2C). After registration, we defined a single set of ROIs across the two recordings and manually checked the quality of each ROI for size, shape, and brightness to identify ROIs that contained the same cell across conditions (Fig. 2D). We found that many neurons, though not all of them, exhibit similar tuning across the two conditions, with small differences in preferred direction (Fig. 2E). Across all recordings, matched ROIs on average have the same heading preference with respect to wall pattern across conditions, suggesting that cells accurately represent heading with respect to the chamber's visual cues irrespective of head or chamber rotation ($V_{32} = 14.1$, $p = 2.5 \times 10^{-4}$, V-test for nonuniformity against 0°; Fig. 2F). Not all cells retain the same heading preference, possibly due to distal visual cues past the walls of the chamber, which are not informative in the chamber rotation condition, but provide potential landmarks in the head rotation condition.

### Changing visual cues elicits coherent remapping in RSC neurons

The previous experiment demonstrates that many of the neurons we found that are tuned to changes in head angle are also tuned for changes in the rotation of the environment. Next, we wanted to

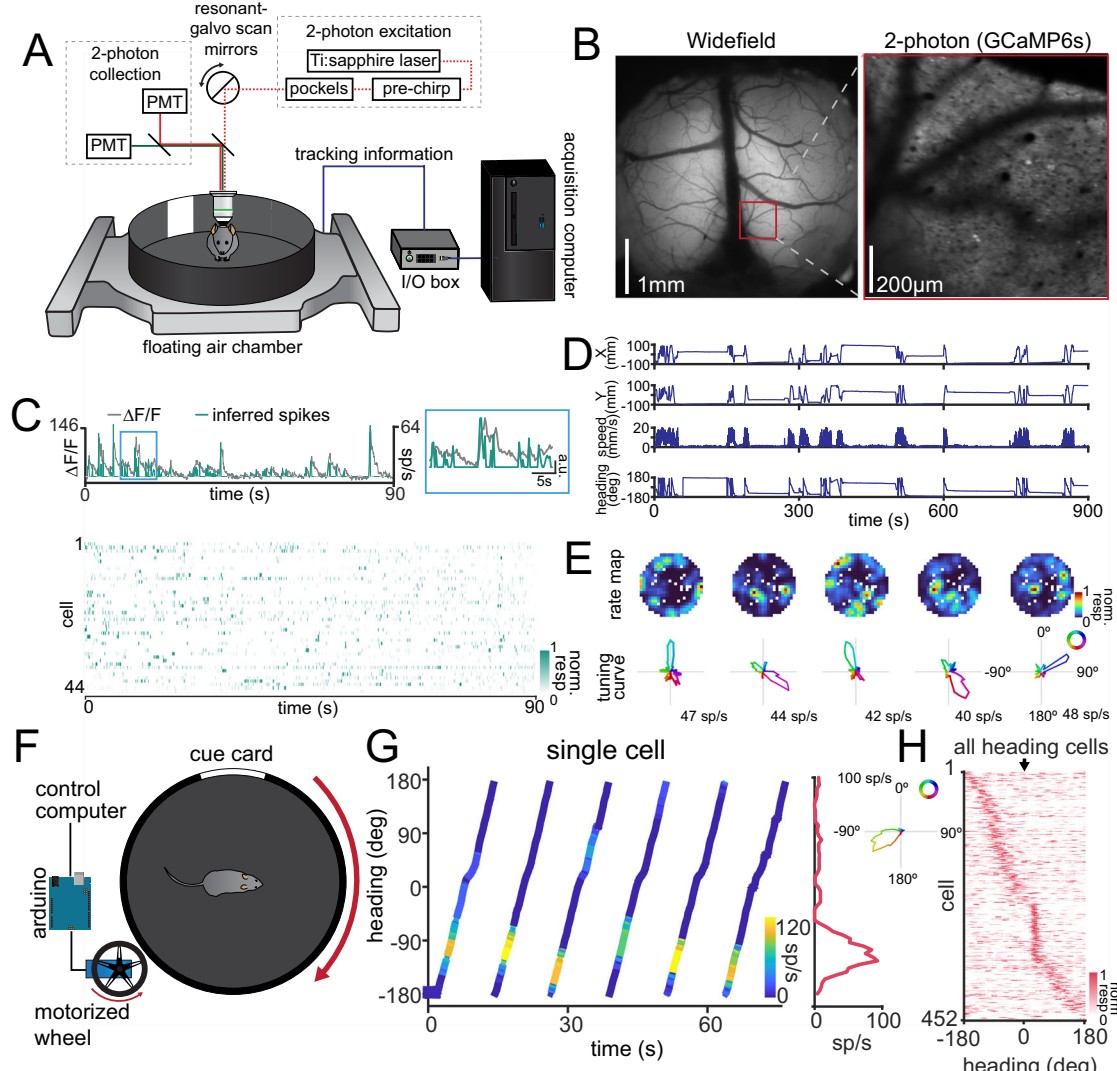

**Fig. 1 | Individual neurons in the RSC exhibit heading responses in the absence of physical head movement. A** Schematic of the imaging set-up. Mice are head fixed in a floating chamber and allowed to voluntarily move the chamber floor using their paws, allowing control of their position and orientation within the chamber. Tracking information from the chamber is synchronized to the collection of the two-photon image acquisition. **B** Left: Example widefield image of the entire cortical window with the location of the imaged plane in dysgranular RSC indicated with a red box. Scale bar = 1 mm. Right: Example imaging plane in RSC using two-photon microscopy (n = 4 imaging fields over 3 mice) Scale bar = 200 μm. **C** Top: Neural responses are first extracted as changes in fluorescence (ΔF/F), then spikes are inferred from the calcium traces. Gray traces show the ΔF/F trace and the green trace shows the inferred spikes. Inset: Zoomed section of the blue box on the left. Bottom: Example inferred spikes from a population of cells in a single recording. **D** Example chamber tracking data from a single experiment, showing X position, Y position, speed, and heading angle. **E** Example rate maps (top) and heading

direction tuning curves (bottom) from neurons recorded from RSC. Note that neurons exhibit selectivity to heading, but not to allocentric position in the floating chamber. **F** Schematic of the controlled rotation set-up. A motorized wheel is added to control the rotation of the floating cage. The mouse is head-fixed in the center of the chamber and the X-Y translation of the chamber is restricted by guide bearings. The red arrows show the rotation of the wheel and chamber. **G** Left: Example of a single neuron's response to rotation in the chamber. The orientation of the chamber ("heading") is plotted against time, and the activity represented by the line color as indicated in the inset colorbar. The neuron consistently responds at the same heading across trials. Right: Average tuning curve across all trials, showing the elevated response of the neuron at a specific heading. Inset: The same tuning curve plotted in polar coordinates. **H** Tuning curves of all responsive neurons, sorted by cross-validated angle of peak response, showing that the preferred headings of the population span the entire environment.

confirm that heading-tuned neurons observed during head rotation exhibit similar population dynamics as HD cells recorded during free movement. HD cells have been found to remap in different environments, while maintaining a constant angular offset between pairs of HD cells. As a consequence, moving an animal to an environment with new visual landmarks elicits "coherent remapping"[59], as the entire population rotates to align to the new landmarks, while retaining the same relative offsets between neurons (Fig. 3A). To test whether heading cells in the rotating environment also have this quality, we replaced all of the visual cues on the chamber wall between recordings

to alter the local landmarks and elicit remapping, creating two separate contexts. When comparing the same cells across the two contexts, we found that the majority of heading cells indeed shift by the same angular offset (n = 12 recordings over 4 mice; Fig. 3B). The angular offset is not consistent across different recordings because the remapping due to changing contexts is random. Therefore, to compare across recordings, we first calculated the "phase offset" of each neuron, defined as the difference in preferred heading across conditions. Then, iterating through each neuron, we subtracted each neuron's phase offset to the averaged phase offset of all other neurons in

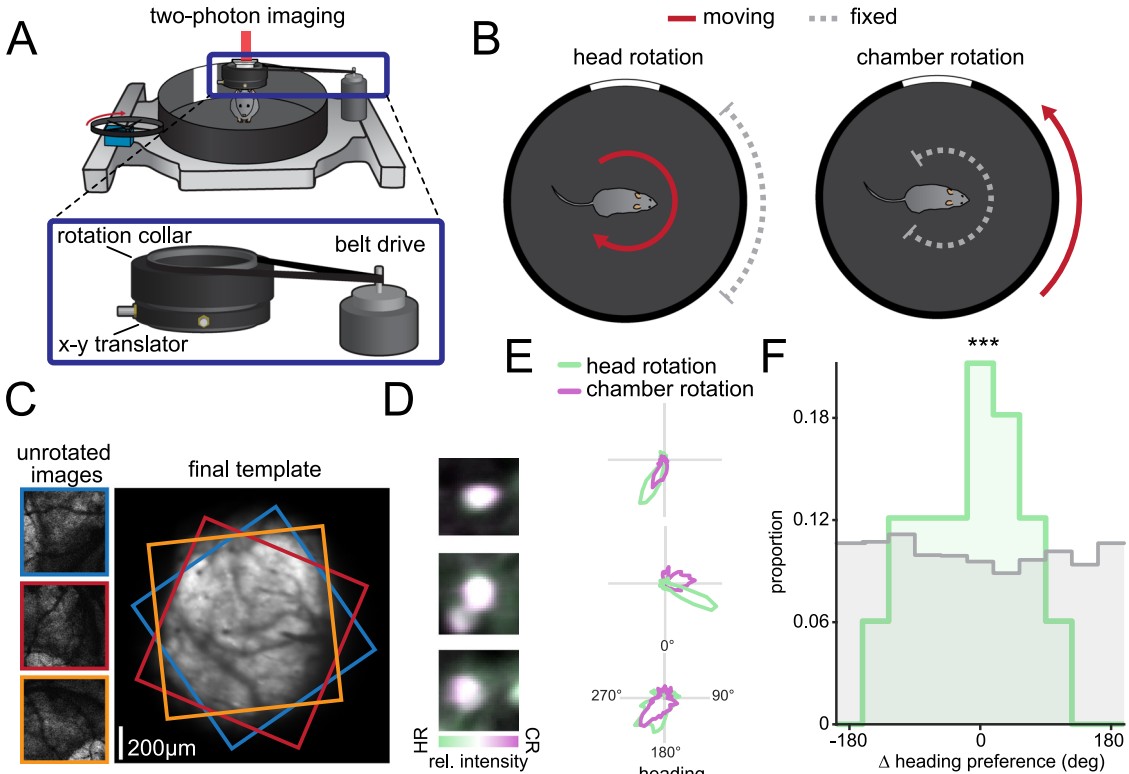

**Fig. 2 | Neurons in the RSC retain similar tuning in head-rotation or chamber-rotation conditions. A** Example of the adapted head-rotation set-up for comparisons between head-rotation and chamber rotation experiments. In addition to the motor-driven wheel that can rotate the chamber while the head is immobilized, there is a separate rotation collar that can be used to rotate the mouse over the stationary chamber. Inset: Zoomed in image of the rotation collar assembly. The rotation collar contains an x-y translator to shift the axis of rotation to the center of the imaging field. A belt-drive connects the collar to a motor for controlled rotation. **B** Schematic of the two experimental conditions. Either the chamber or the head rotates independently, shown with a red arrow, while the other remains stationary, shown with a dashed gray capped line. In both cases, this causes the visual cue card to move across the visual field in the same direction. **C** Example of derotation process for image timeseries. Each of the three raw images is rotated to match the template on the right. Colored borders show the position and rotation of each individual image. The final template shown on the right is after all the images have been registered for the recording ($n = 5$ imaging fields over 4 mice). **D** Example of three matched ROIs from the recordings. Purple denotes greater brightness in the chamber rotation recording, whereas green denotes greater brightness in the head-rotation recording. White denotes equal brightness and suggests good alignment of the ROIs across recordings. HR: Heading rotation, CR: chamber rotation. **E** Three example tuning curves from pairs of ROIs showing similar tuning preferences. The tuning curve in the head rotation condition is shown in green, whereas the tuning curve in the chamber rotation condition is shown in purple. **F** Histogram showing the difference in peak heading preference across recordings for matched ROIs. The zero-centered peak in the data (green) indicates that cells remained tuned to the similar heading angles independent of head- or chamber-rotation. Shuffled distribution is shown in gray. ***$p = 2.5 \times 10^{-4}$, V-test for circular nonuniformity against 0°.

the same recording ("Δ phase offset"; see "Methods") to determine whether the neuron coherently remapped along with the population. When combining measurements across all recordings, we found that changing contexts caused the population of heading cells to coherently remap, as indicated by low Δ phase offsets ($V_{66} = 25.6$, $p = 4.9 \times 10^{-6}$, V-test for nonuniformity against 0°; Fig. 3C).

In summary, we show that heading cells retain their tuning whether vestibular signals are modulated (in head rotation) or not modulated (in chamber rotation). In addition, we show that the heading network in a head-fixed preparation exhibits the same functional properties as in freely moving animals. Together, these results show that modulation of vestibular input is not necessary for eliciting heading responses in RSC if vision and proprioception are spared. Rather, many of the same heading neurons remain active and accurately represent the animal's heading relative to the moving environment. Importantly, these results also show that the mammalian heading network is capable of representing the animal's heading relative to the environment even when the head is immobilized.

### The ADN and visual cortex send distinct information to RSC

In the previous experiments, we found that not all cells would retain their preferred direction across conditions or coherently remap,

suggesting that there is significant heterogeneity in RSC, which may be due to differences in the inputs it receives from other brain regions. Previous anatomical tracing studies[38,40] have shown that RSC receives direct projections from both ADN and the visual cortex which likely contribute to RSC heading representations. However, the responses of these projections have not been studied in behaving animals, and the information that each area sends to RSC is not known. To measure the responses from RSC-projecting neurons in each of these regions, we microinjected an AAV expressing GCaMP7b into ADN or the higher visual areas anteromedial (AM) and posteromedial (PM) in separate cohorts of wild-type mice. We chose AM and PM since they provide the major input from visual cortex to RSC[60] as well as their role in visual processing[61]. We then imaged the axon terminals arising from ADN ($n = 17$ recordings over 4 mice) or AM/PM in RSC ($n = 7$ recordings over 3 mice; Fig. 4A). In order to distinguish classical visual responses from heading responses, we used a pair of symmetric cues on opposite sides of the chamber so that an identical visual stimulus would pass in front of the mouse twice for each full chamber rotation. We also recorded in both light-on and light-off conditions to selectively gate visual input (Fig. 4B). We postulated that projections carrying heading signals would exhibit unimodal peaks that persist in the light off condition, accurately representing heading regardless of light condition or cue

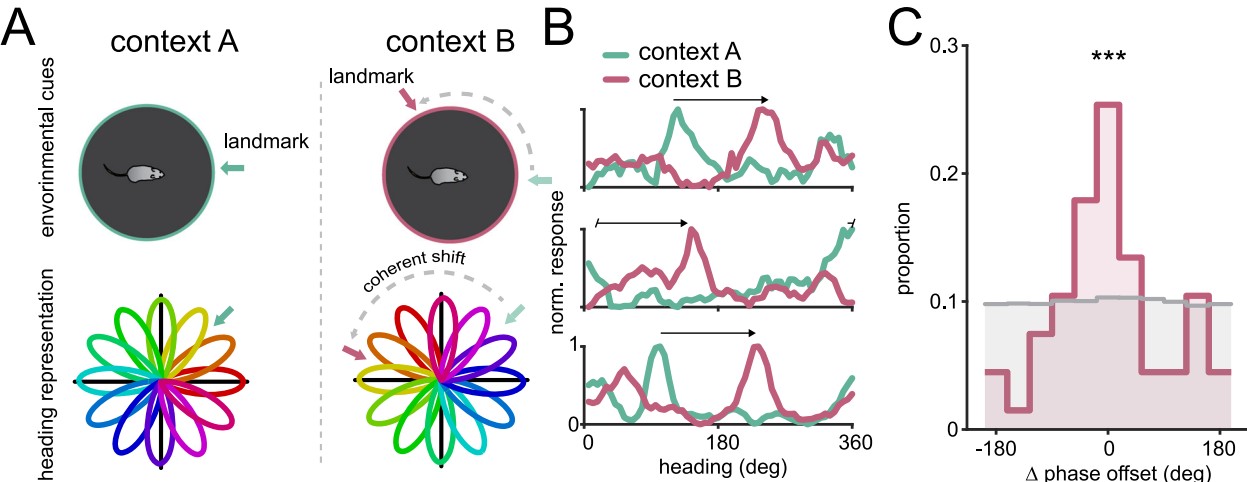

**Fig. 3 | The heading network coherently remaps when visual cues are changed.**
**A** Schematic showing coherent remapping across two different contexts. Because each individual neuron's preferred heading is mapped as a relative offset to visual landmarks, the relationships between neuron offsets would be expected to be preserved across contexts, leading to a coherent population shift. **B** Three example neurons from the same session showing a unified shift in preferred heading from context A to context B. Tuning curve from context A is shown in green, while tuning curve from context B is shown in purple. Arrow shows the phase offset between contexts. **C** Histogram showing that neurons coherently remap across conditions. For each recording session, a neuron's coherence with the population was calculated as the difference between the neuron's degree of remapping and averaged degree of remapping across the remainder of the population (see "Methods"). The peak around 0 suggests in the real data (purple) shows that individual neurons are shifting preferred directions in a similar amount to the rest of the population in each recording. Shuffled distribution is shown in gray. ***$p = 4.9 \times 10^{-6}$, V-test for circular nonuniformity against 0°.

symmetry. On the other hand, projections communicating visual landmark signals would exhibit a light-sensitive and bimodal tuning curve, with peaks separated by approximately 180° due to the symmetry of the visual stimulus (Fig. 4C).

We found that responses from ADN and visual cortical projections exhibit distinct responses in these conditions. In the light-on condition, ADN axons show a unimodal tuning curve, whereas visual cortical axons show a bimodal tuning curve (Fig. 4D, E). In the light-off condition, ADN axons maintain their response for several trials, whereas visual cortical axons lose their tuning. These results show that the information from the visual cortex is highly sensitive to environmental cues and cannot be entirely sufficient for driving heading responses, as any symmetrical environment would result in an inaccurate heading representation. The accurate internal heading representation regardless of environmental symmetry suggests that the information from the visual cortex must be processed prior to being integrated into the heading network. Together, these results indicate that ADN and visual cortex send distinctively tuned projections to RSC, which provide a basis for aligning the heading signal with external cues.

### A functional class of RSC neurons coregisters visual and heading signals

How do neurons in RSC combine the disparate responses from ADN and visual cortex? We imaged calcium activity in RSC cell bodies using the same experimental conditions, allowing us to determine whether RSC responses resemble thalamic or visual cortical inputs ($n = 36$ recordings over 6 mice). We found a diversity of responses in RSC that resembled both visual cortical inputs (discernible by a second peak in light) and ADN inputs (discernible by a single peak in the dark) (Fig. 5A, Figure S4A-B). These responses persisted regardless of rotation direction (Figure S5, $p = 0.0094$, bootstrapped KS-test, $n = 4$ recordings over 2 mice) or whether or not the animal was in control of the arena (Figure S6, 0.031, bootstrapped KS-test, $n = 7$ recordings over 3 mice). To help classify RSC responses, we fit a sum of Gaussians to the data then performed unsupervised clustering on the coefficients using a Gaussian mixture model (GMM) to separate the cells into functional classes (Fig. 5B; see "Methods"). We found that the data were optimally clustered into three functional classes (see "Methods"), although we

note that the responses span a continuum rather than fully isolated clusters.

The model returned a prominent class of cells that mirrored the tuning curves of ADN axons, with a unimodal peak that is stable in either light-on or light-off conditions, which we call "heading cells" (Fig. 5C). We also found another class that mirrored the tuning curves of visual cortical axons, with a bimodal peak in the light-on condition that disappears in the light-off condition, which we call "landmark cells" (Fig. 5D). Finally, the clustering returned a third major class of cells, which combined the responses of ADN and visual axons, exhibiting bimodal responses in the light-on condition, but changing to a more unimodal response profile in the light-off condition, which we call "alignment cells" (Fig. 5E). The light-off peak in these alignment cells was always at the same location as one of the two peaks in the light-on condition, suggesting that landmark and heading responses are precisely coregistered in the alignment cells. Of the total number of tuned cells ($12.1 \pm 1.1\%$), landmark cells made up the plurality of cells ($33 \pm 3.5\%$), followed by alignment cells ($28 \pm 1.9\%$, mean ± s.e.m.), and heading ($22 \pm 2.3\%$) cells. We then compared the flip scores in the light-on and light-off conditions across cell classes[46]. The flip score measures the bimodality of a neuron's tuning curve, providing a direct measurement of the sensitivity of the neuron's responses to the symmetric visual cues. Heading cells showed no significant difference in flip score across conditions ($F_{1,662} = 0.18$, $p = 0.67$, Fig. 5F). On the other hand, both landmark ($F_{1,1042} = 1062.5$, $p = 3.1 \times 10^{-161}$) and alignment ($F_{1,816} = 209.5$, $p = 2.0 \times 10^{-42}$) cells have significantly lower flip scores in the light-off than the light-on condition. These results suggest that there are separate functional clusters present in RSC population which are differentially influenced by visual information.

Since ADN and visual cortex send tuned projections to RSC, we hypothesized that the somatic responses would be the result of specific combinations of tuned connections. To show this, we fit gaussians to the axonal data using the same methods as for the somatic data. We then plotted them using the same linear discriminant axes, finding that ADN axons and visual axons are mostly restricted to the boundaries of the heading and landmark clusters, respectively (Fig. 5I), although there is some overlap with the alignment cluster boundary (Figure S7). To further confirm this, we compared neurons in the standard rotation

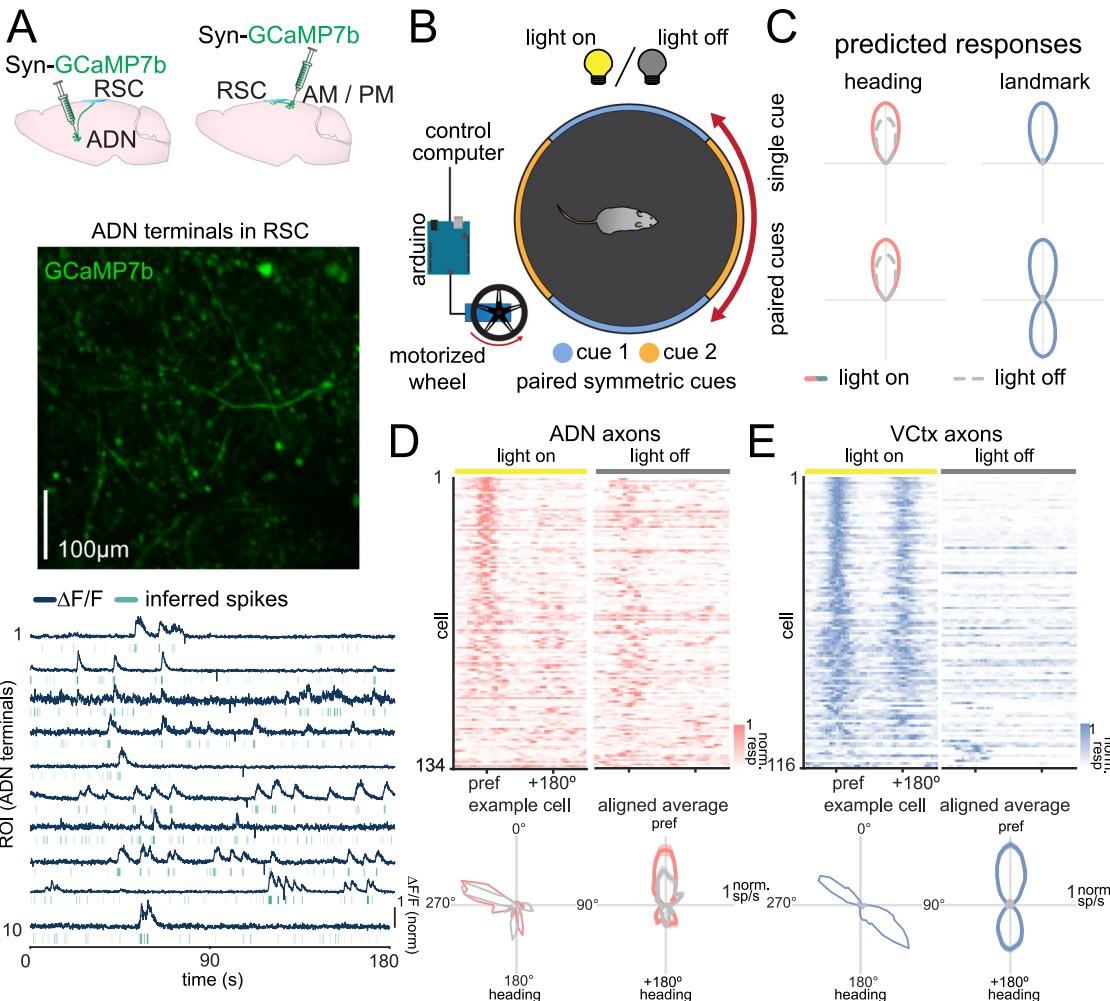

**Fig. 4 | Imaging projections from ADN and visual cortex reveal distinctly tuned responses sent to RSC. A** Top: Example of injection sites in separate cohorts of mice. AAV-Syn-GCaMP7b was injected into either ADN or AM/PM of WT mice. Middle: Example imaging field in RSC of axon terminals originating from ADN (*n* = 17 imaging fields over 4 mice). Bottom: Example Δ*F/F* traces and inferred spikes for imaged ADN terminals it RSC. **B** Example schematic of experimental conditions. The single white cue card has been replaced with paired symmetric cues on the floating chamber, which is driven by a motorized wheel. Neurons from RSC are recorded across light-on and light-off phases. **C** Schematic of expected axonal responses. Left: Axons terminals encoding heading independent of visual cues are expected to have unimodal tuning curves across both conditions that are not

affected by the light condition. Right: Axons terminals responding to visual cues are expected to show a single peak in a single cue condition, but dual peaks in the paired symmetric cue condition. In both cases, the responses will be reduced in the light-off condition. **D** Responses of ADN axons across recordings. Top: Heat maps in the light-on (left) and light-off (right) condition. Each row represents the aligned response of a single axon, and axons are organized by similarity to a group mean in the light-on condition. Bottom: Single axon response in light-on and light-off conditions (left). Averaged response across all ADN axons, showing an aligned unimodal peak in both light-on and light-off conditions (right). Traces shown are mean ± s.e.m. **E** Same as (**D**), but for visual cortex axons. Averaged response shows a bimodal peak in the light-on condition, and no response in the light-off condition.

condition with visual cue rotation only (Fig. 5J). We found that the visual contributions to tuning were retained in visual cue only rotation, but the loss of proprioceptive and somatosensory information strongly degraded the heading cell responses (Fig. 5K). These results support that projections from visual cortex and ADN innervate specific targets in RSC, leading to a functional class of cells capable of cor-egistering visual cues and heading.

**Contributions of distinct classes to representation of heading**
The functional classes observed would be expected to lead to different representations of heading direction. We sought to understand the contributions of each functional class to the population-level coding of heading. First, we confirmed that the heading network was able to accurately represent the animal's heading, even in an environment with symmetric visual information, by decoding heading from popu-lations of cells within a single imaging frame. To combine multiple recordings, we created pseudopopulations of neurons by aligning

trials across recordings and then randomly sampling a subset of these cells, determined by the number of cells in the smallest class (see "Methods"). Restricting decoding to subpopulations of cells within single imaging frames (of 100 ms duration) increased the dynamic range in decoder performance and allowed us to directly compare performance between classes. In the light-on condition, there is a slight systematic overrepresentation of 180° errors due to the symmetric visual cues. To prevent 180° decoding errors from biasing the per-formance metric, we opted to measure performance using "decoder accuracy", which is defined as the fraction of decoded headings that are within 18° of the actual heading in each time bin. When sampling across all the classes, the heading network shows remarkable accuracy, even in the presence of symmetric visual cues (mean decoder accuracy = 0.82, 95% CI = [0.78, 0.86], Fig. 6A). When transitioning from light-on to light-off, there is a drop in the performance of the decoder, although the performance remains above chance, suggesting that the heading network continues to track heading, albeit less accurately, even in the

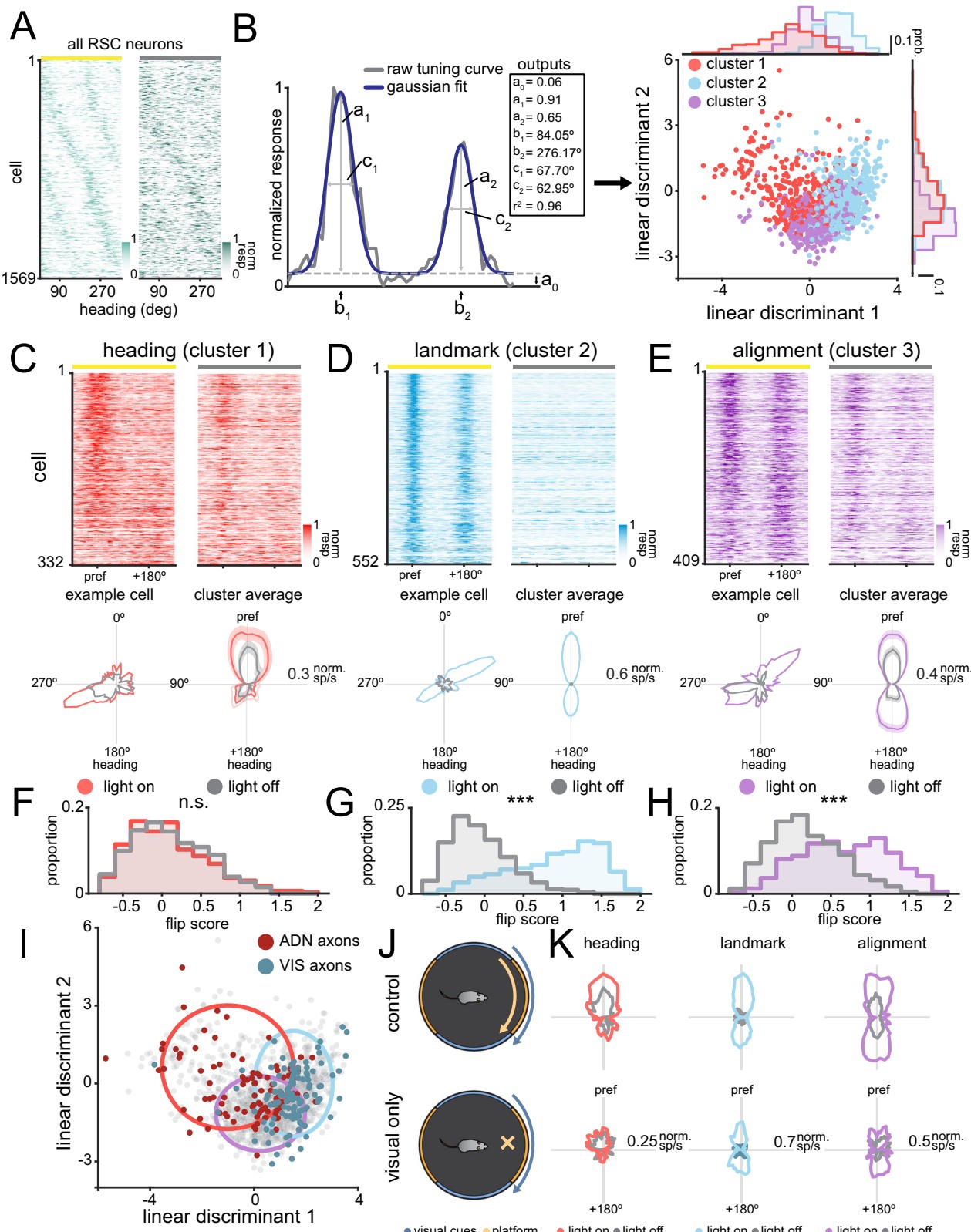

absence of visual cues (mean decoder accuracy = 0.44, 95% CI = [0.34, 0.54], $p = 7.6 \times 10^{-3}$; Fig. 6A, bottom). To further parse the contributions of specific functional classes, we also performed population decoding restricted to cells of one response type (heading, landmark, alignment; Fig. 6B). As with the entire pseudopopulation, we can examine the differences in decoder performance in light-on vs light-off conditions as well as the contribution of the symmetric visual cues,

which manifests as increased 180° errors. We found that heading cells were able to accurately decode in both light-on (mean decoder accuracy = 0.61, 95% CI = [0.56, 0.66]) and light-off conditions (mean decoder accuracy = 0.44, 95% CI = [0.36, 0.52]), suggesting that they contribute to the heading signal in both conditions ($p = 0.11$; Fig. 6B, top). Since their decoder performance is lower than the population in the light-on condition, heading cells alone may be insufficient for

**Fig. 5 | Unsupervised clustering of neural responses in the RSC reveals distinct functional classes. A** Heat maps of the responses of all somas imaged in RSC. Each row corresponds to a neuron in the light-on and light-off condition, and responses are ordered by their preferred direction. Light-off responses are ordered based on their light-on position. **B** Left: Example of a two-term Gaussian fit on an aligned and normalized tuning curve. The outputs of the two-term Gaussian for light-on and light-off conditions are used for clustering. Right: Scatter plot of clustered responses in a reduced dimension (LDA space). Histograms along the edges show the distribution of values along each dimension. Cluster numbers are denoted by color. Note that LDA plot is only for visualization, clustering was performed in full 8-dimensional space (see "Methods"). **C** Top: Heat maps showing the aligned responses of the heading neurons (cluster 1), ordered by similarity to a cluster averaged response. Bottom left: Example tuning curve in light-on and light-off

conditions for a single cell. Bottom right: Averaged responses of the neurons in the heading cluster in light-on and light-off conditions. Traces are shown as mean ± s.e.m. **D** Same as (**C**), but for landmark neurons (cluster 2). **E** Same as (**C**), but for alignment neurons (cluster 3). **F** Histogram showing light-on and light-off flip scores for the heading cluster. $p = 0.67$, F-test. **G** Same as (**F**), but for landmark neurons (cluster 2). $p = 3.1 \times 10^{-161}$, F-test. **H** Same as (**F**), but for alignment neurons (cluster 3). $p = 2.0 \times 10^{-42}$, F-test. **I** Location of axonal data on the reduced dimensional space. Gray dots are the positions of RSC soma. Red dots are ADN axons and blue dots are visual axons. Colored circles represent the 95% CI ellipses surrounding each cluster, with colors as above. **J** Schematic of standard rotational experiment versus visual cue only rotation. **K** Light-on and light-off tuning curves of matched neurons in the standard rotation (top) and visual cue only rotation (bottom) for each cell cluster.

providing an optimal representation of instantaneous heading (Fig. 6C, right; Table S1). As expected based on their tuning curves, heading cells did not exhibit a preponderance of 180° errors, suggesting they do not represent visual information directly and were therefore not affected by the symmetric cues. In contrast, landmark cells have very high decoder performance in the light-on condition (mean decoder accuracy = 0.80, 95% CI = [0.75, 0.84]), with an overrepresentation of decoder errors at 180°, which immediately plummets to chance levels in the light-off condition (mean decoder accuracy = 0.20, 95% CI = [0.14, 0.27]; $p = 8.0 \times 10^{-4}$; Fig. 6B, middle). This suggests that landmark cells strongly drive the heading signal when visual cues are present but provide almost no contribution in their absence (Fig. 6C, right; Table S1). Lastly, alignment cells show decoder performance that is between that of heading and landmark cells, taking advantage of improved decoding from the visual cues in the light-on condition (mean decoder accuracy = 0.71, 95% CI = [0.66, 0.76]), but retaining fairly accurate heading in the light-off condition (mean decoder accuracy = 0.51, 95% CI = [0.43, 0.59], $p = 0.54$; Fig. 6B, bottom). This suggests that alignment cells can take advantage of the visual cues in the light-on condition, while retaining accurate decoding in their absence in the light-off condition, positing that they may be responsible for integrating and registering visual landmark and heading information (Fig. 6C, right; Table S1).

Taken together, we propose a working model to describe the distinct roles played by heading, landmark, and alignment cells in the registration of heading representations to visual cues (Fig. 6C). First, in the light-on condition, all three classes are active, with landmark cells faithfully representing the position of the visual cues, heading cells relaying information from the heading network, and a subset of alignment cells coregistering both sources of information, resulting in an internal heading that is well aligned with the true external heading (Fig. 6C, left). In the light-off condition, the landmark cells are silenced, leaving the heading cells as the primary driver of internal heading, and the alignment class exhibiting a response matched to the heading cells (Fig. 6C, center). Moreover, the lack of anchoring visual information causes the heading network to drift over time, and the internal heading to becomes offset from the true heading. Upon restoring visual cues in the light-on condition, landmark cells regain their tuning, relaying this information to the appropriate subset of alignment cells, which are in turn updated (Fig. 6C, right). The alignment cells then update the heading cells, and the broader network via feedback projections from RSC to the ADN, updating the heading representation throughout the heading system. We propose that the co-registration of visual landmark and heading signals in the alignment cells thereby plays a critical role in aligning the heading representation to external visual cues.

## Discussion

One of the key findings of this paper is that heading, the relative orientation of an animal in an environment, can be encoded separately from physical head direction in the heading network. This finding is consistent with previous studies in which humans solve complex

virtual navigation tasks without the requirement of physical movement, suggesting that they can create an abstract mental map of their direction independent of physical locomotion[62–65]. Research in invertebrates further supports this view, as the E-PG ("compass") neurons of the *Drosophila* central complex, an analog of the heading network, are active in virtual environments even when the fly is immobile[1,2,31].

Here, we demonstrate for the first time that neurons in the mammalian heading network show a similar capability during head immobilization, using multiple sensory inputs to accurately represent relative heading. Although previous experiments have examined the influence of non-vestibular sensory inputs on heading, they have uniformly allowed free angular rotation of the head[20–23]. Indeed, previous experiments with head-fixed mice also allowed for rotation of the animal's head, either via a rotational mount[66] or physical rotation of the entire animal[67]. As compared to traditional head-fixed virtual reality setups, our floating chamber designs confers several benefits. Beyond providing a three-dimensional visual environment, the floating chamber maintains somatosensory and proprioceptive cues present during locomotion through a physical environment. Using a combination of these sensory cues, the heading cells (many of which would be classified as HD cells in freely moving conditions) appear to be able to compensate for the absence of vestibular information, although not all neurons maintained their tuning between head rotation and chamber rotation conditions. Because of the heterogeneity of responses in RSC, a subset of RSC heading cells may be more strongly driven by vestibular input, and therefore lose their response during head fixation; whereas more visually driven neurons are still responsive. This is reminiscent of previous studies which found that including additional sensory modalities improves the coding of AHV[51,68] and HD[35] cells. However, our results show that, despite the reduced activity due to the absence of vestibular input, neurons of the heading network in head-fixed mice remain functional and are able to accurately track heading, expanding the repertoire of experimental approaches.

The HD network has commonly been modeled as a ring attractor network, with heading represented by neurons as an "activity bump" that can be shifted by sensory inputs. Along with the results presented here, other recent work has begun to elucidate the specific cells and mechanisms that may be responsible for anchoring the activity bump to visual landmarks. Jacob et al. found a specific class of cells that have bimodal tuning curves in a two-chamber symmetric environment, called "within-compartment bidirectional cells", that were postulated to develop their responses through Hebbian-based strengthening of visual information with information from the heading network[45,46]. Similarly, LaChance et al.[69] found bidirectionally tuned cells in the postrhinal cortex that could represent and discriminate between symmetric visual cues, suggesting an important role in processing visual landmarks. Lastly, recent modeling work[70,71] suggests mechanisms within the RSC that may be specifically responsible for anchoring heading to landmarks. However, there are some key differences in our findings. Previous work in RSC has found bimodal responses in two

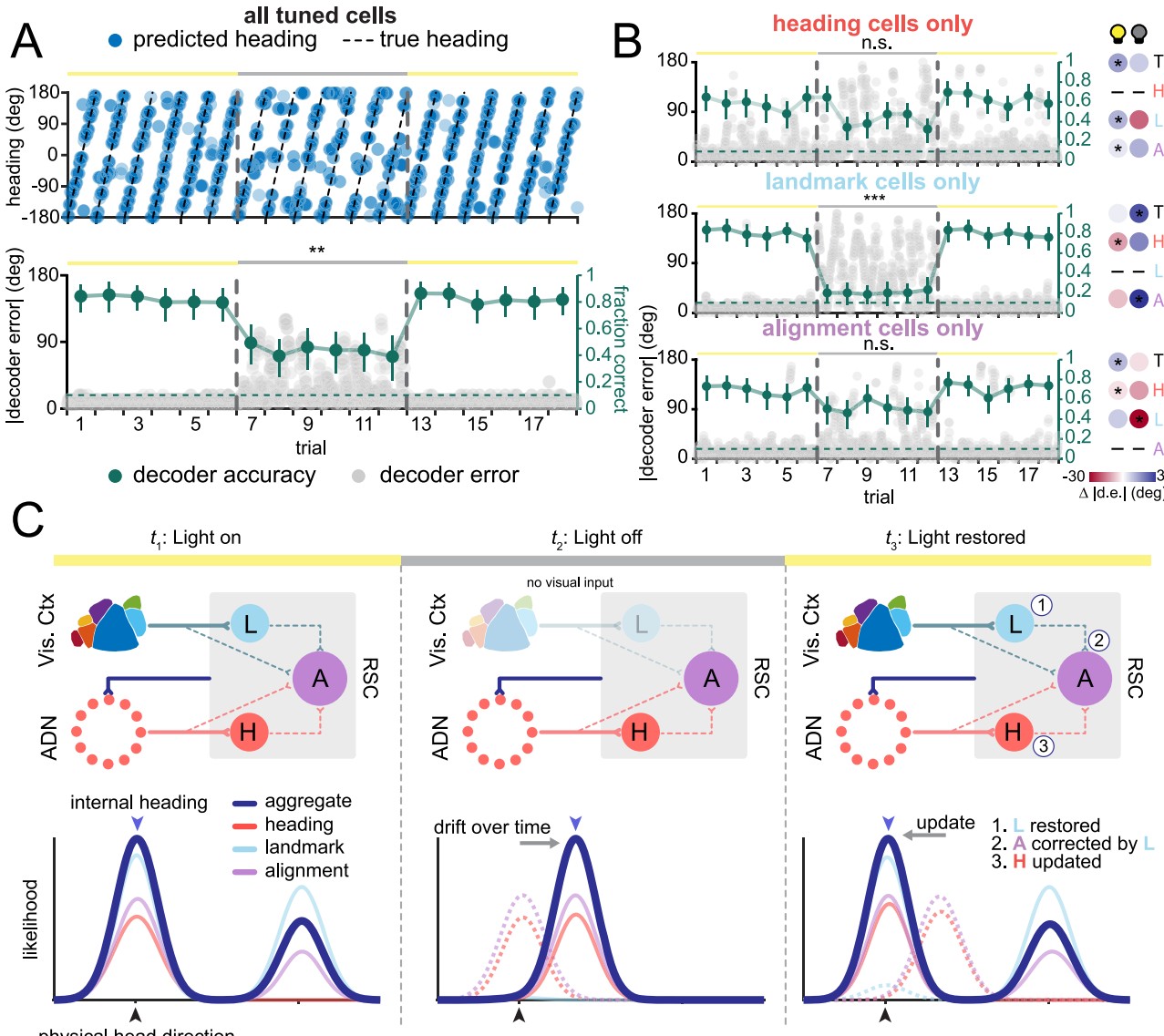

**Fig. 6 | Decoding analysis shows independent contributions of each functional subclass to the overall heading representation in RSC. A** Pseudopopulation decoding of neural responses across trials. Top: Predicted heading plotted for each 100 ms time bin (blue dots) overlaid on actual heading (black dotted line) for each trial, transitioning from the light-on (yellow bar) to the light-off (gray bar) condition. The transition line between conditions is shown as a vertical gray dashed line. Bottom: Decoder error (gray dots) and the average decoder accuracy for each trial (green dots and line). The dashed green line denotes chance level for decoder accuracy (0.10). Decoder accuracy is plotted as mean ± s.e.m. Bootstrapped t-test (see "Methods"), $n = 1000$ bootstrapped samples. $p = 7.6 \times 10^{-3}$. **B** Top: Decoder error (gray dots) and decoder accuracy (green dots and line) for decoder using only heading cells across conditions (left). $p = 0.11$ Comparison of decoder error between heading class cells and other cell classes (T: total, H: heading, L: landmark, A: alignment) showing differences in performance across classes (right). Red shading indicates that the heading class outperforms the compared class; blue shading indicates the opposite. Middle: Same as top, but for landmark cells only. $p = 8.0 \times 10^{-4}$. Bottom: Same as top, but for alignment cells only. $p = 0.54$ Decoder accuracy is plotted as mean ± s.e.m. Bootstrapped t-test (see "Methods"), $n = 1000$ bootstrapped samples. **C** Schematic of circuit for integration of visual information into the HD network, indicating changes in circuit representation during light-on condition (left), light-off condition (middle), and back in the light-on transition (right). Top: The blue feedback line describes putative feedback from the RSC to the ADN, sending adjusted heading information. Bottom: Population likelihood curves are shown in dark blue, with each cell class (heading, landmark, and alignment) shown in red, blue, and purple, respectively. Blue arrows above the likelihood curves denote the predicted internal heading at each time point. Black arrows below the likelihood curves denote the physical head direction at each time point. Dotted lines denote the previous position of the likelihood curve. n.s.: not significant, *$p < 0.5$, **$p < 0.01$, ***$p < 0.001$. Linear mixed-effects model.

connected visually symmetric chambers, but not in a single chamber with symmetric cues[72]. In our recordings, we found cells in RSC with bimodal tuning curves in a single chamber with symmetric cues, akin to studies in postrhinal cortex[69]. We postulate that the difference in bimodal responses in RSC may be due to the mouse being fixed in the center, removing modulation of both vestibular signals and proximity-dependent visual cue size. Although our results are complementary to this previous work, experimentally dissociating visual and heading

contributions enables the separation of cell types that were previously combined. For example, although landmark and alignment cells may have similarly bidirectional responses in symmetric environments, removing visual input affects their responses very differently. As a result, our experiments reveal novel cellular responses which support the integration of visual and heading information.

Our results show that RSC integrates visual information into the heading network, but other areas in the HD network may also play

important roles. The postsubiculum shares many of the same properties as RSC, as the only other area in the heading network receiving substantial monosynaptic input from both visual cortex and ADN[73,74]. Lesioning postsubiculum degrades the heading signal in the ADN and results in an inability to register to visual landmarks, similar to RSC lesions[75]. Recent work has also suggested that the postrhinal cortex may also play a role, as it contains visually driven heading cells that can discriminate between symmetric landmarks[69]. In addition, cells with bimodal responses have also been found in the medial entorhinal cortex[76], which has been shown to affect RSC activity[77]. Given that cells in each of these areas exhibit similar visually-influenced tuning properties, are these systems redundant or complementary? Since the regions are highly interconnected[40,78], it is possible that they may redundantly represent the same information, since registration of visual information is generally important for maintaining heading during spatial processing. Alternatively, the processing in each area may be complementary, biased by the strength of their connectivity with other brain regions[43]. For example, different visual areas may provide different information (e.g., landmarks, coherent motion) to downstream structures, which can integrate the diverse visual inputs with heading direction signals. Further studies are necessary to understand how the distributed regions in the HD cell network interact during navigation.

In conclusion, our study provides evidence that neurons in the HD cell network are capable of representing the heading of the animal relative to the environment independent of physical head direction. The dysgranular RSC is able to combine multisensory information to create an abstracted map of heading, similar to how place cells can encode abstract mappings of non-positional variables[79–81]. Finally, we propose a potential circuit based on a newly discovered class of neurons in RSC that act to coregister visual cues to the heading network. Future studies of how heading and other spatial signals are aligned to sensory input will be critical for understanding how internal representations of the spatial environment are generated, maintained, and updated by sensory information.

## Methods

### Animals
For cortex-wide calcium indicator expression, Emx1-Cre (Jax Stock #005628) × ROSA-LNL-tTA (Jax Stock #011008) × TITL-GCaMP6s (Jax Stock #024104) triple transgenic mice or Slc17a7-IRES2-Cre (Jax Stock #023527) × TITL2-GC6s-ICL-TTA2 (Jax Stock #031562) double transgenic mice were bred to express GCaMP6s in cortical excitatory neurons. For axon imaging experiments, wild-type C57BL/6J mice were used. For all imaging experiments, 6–12-week-old mice of both sexes (12 males and 14 females) were implanted with a head plate and cranial window and imaged starting 2 weeks after recovery from surgical procedures and up to a maximum of 10 months after window implantation. Mice were housed in cages of up to 5 animals prior to the implants, and singly housed post-surgical procedure in a 12:12 light-dark cycle with the following controlled parameters: temperature (68–76 ℉), humidity (30-70%), ventilation (10–15 air changes per hour). All animal procedures were approved by the Institutional Animal Care and Use Committee at UC Santa Barbara.

### Surgical procedures
All surgeries were conducted under isoflurane anesthesia (3.5% induction, 1.5–2.5% maintenance). Prior to incision, the scalp was infiltrated with lidocaine (5 mg kg⁻¹, subcutaneous) for analgesia and meloxicam (2 mg kg⁻¹, subcutaneous) was administered pre-operatively to reduce inflammation. Once anesthetized, the scalp overlying the dorsal skull was sanitized and removed. The periosteum was removed with a scalpel and the skull was abraded with a drill burr to improve adhesion of dental acrylic. For all chamber-rotation mice, a 4 mm craniotomy was made centered over the midline (centered at 3.0 mm posterior to Bregma),

leaving the dura intact. For a subset of these mice, AAV-Syn-Flex-jGCaMP7b (Addgene #104493) was injected into either AM (3.0 mm anterior, 1.7 mm lateral), PM (1.9 mm anterior and 1.6 mm lateral to Lambda), or ADN (0.8 mm posterior and 0.8 mm lateral to Bregma, 3.2 mm depth[82]) of Emx1-Cre mice. For head rotation experiments, the 4 mm craniotomy was centered 3.0 mm posterior and 1.0 mm lateral to Bregma, so that the axis of rotation was not centered over the sagittal sinus. A cranial window was implanted over the craniotomy and sealed first with silicon elastomer (Kwik-Sil, World Precision Instruments) then with dental acrylic (C&B-Metabond, Parkell) mixed with black ink to reduce light transmission. The cranial windows were made of two rounded pieces of coverglass (Warner Instruments) bonded with a UV-cured optical adhesive (Norland, NOA 61). The bottom coverglass (4 mm) fit tightly inside the craniotomy while the top coverglass (5 mm) was bonded to the skull using dental acrylic. A custom-designed stainless-steel head plate (eMachineShop) was then affixed using dental acrylic. In all conditions, the head plate was carefully positioned to ensure a natural gait under recording conditions. After surgery, mice were administered carprofen (5 mg kg⁻¹, oral) every 24 h for 3 days to reduce inflammation. The full specifications and designs for head plate and head fixation hardware can be found on our institutional lab website (https://goard.mcdb.ucsb.edu/resources).

### Floating chamber design and set up
For the floating chamber experiments (Fig. 1A), we used a set up based on the Mobile HomeCage system (Neurotar Ltd.). Briefly, a 250 mm diameter carbon fiber chamber was placed on top of a perforated metal base. Pressurized air fed into the metal base floated the chamber, reducing friction and allowing the chamber to move readily. Mice were then head fixed in the center of this chamber and allowed to ambulate, which moved the chamber around the mouse while the mouse remained still. The air pressure was tuned (2–4 PSI) so that the chamber provided enough friction to create a natural gait. The chamber contains a set of magnets which are read by the base in order to report information about the position of the chamber (Tracking Software v2.2.1, Neurotar Ltd.), which are then transformed using custom code (MATLAB 2020b, Mathworks) to extract the position, angle, and velocity of the mouse relative to the chamber.

For single cue experiments, the walls of the chamber contained a single black cue (width: 60 mm) set on top of a white noise background that spanned the entire inner circumference of the chamber.

For dual cue experiments, the walls of the chamber contained a repeating pattern of stars or bars (width: 180 mm), in an A–B–A–B pattern. These stimuli were separated by black bars to separate the cues (width: 15 mm).

For remapping experiments, the walls of the chamber were divided into four quadrants, each with a different stimulus. Between recordings, the stimuli were replaced with four new unique stimuli in complete darkness, so that the mouse could not see the transition between stimuli.

All wall cues were printed on nylon waterproof paper and attached to the chamber walls using double-sided tape.

**Forced rotation experiments.** To ensure even sampling of the entire rotation space, we carefully controlled the rotation of the chamber. Three stabilizing bearings (608-2RS) were mounted on optical posts (TR-series, Thorlabs) and placed around the cage to eliminate any X-Y translation. A DC motor (ZGA37RG, Greartisan) was coupled to a rubber wheel and used to spin the chamber using the outer surface. The speed of the DC motor was controlled with a microcontroller (UNO R3, Arduino) drive the rotation of the chamber (3–7 rpm).

The chamber was rotated for 90–120 s followed by a 5 s rest period, which was then repeated 2–6 times for each recording. Mice were monitored via video to ensure that they followed along with the chamber rotation. The total time spent rotating on the floating air

chamber never exceeded 15 min, in order to limit potential discomfort.

For experiments in light and darkness, the lights in the experimental box were switched using a microcontroller (UNO R3, Arduino). The initial repeat was always performed in a light on trial, so that the mouse could register visual landmarks in its environment. Afterward, the box lights were turned on or off in between repeats, during the rest period.

The chamber was thoroughly cleaned and disinfected between experiments in order to remove any odor traces.

**Physical head rotation experiments.** The head rotation apparatus consisted of a rotation collar (LCRM2, Thorlabs Inc.) attached to a translating mount (LM2XY, Thorlabs Inc.) and a quick release adapter (SM2QA, Thorlabs Inc.) The translating mount allowed for independent adjustment of the imaging field about the axis of rotation to optimize the imaging field. The quick release adapter was used in order to easily head fix mice to the apparatus. A closed timing belt for 3D printers (2GT-610) was looped around the rotation collar and attached to a DC motor (ZGHA37RG, Greartisan) and belt tensioner (6 mm belt pulley). Due to the danger of physically rotating the mouse, we limited the speed of the head rotation to 2 rpm and installed safety bars alongside the body of the mouse to ensure that the body of the mouse did not become twisted under the collar.

**Visual cue only rotation experiments.** To rotate the visual cues independently of the chamber floor, we suspended a rigid nylon sheet underneath the mouse and just above the floor of the rotating chamber. This false floor was separated from the rotating chamber so that the rotation of the chamber walls did not cause any movement of the floor, eliminating somatosensory and proprioceptive cues.

## Two-photon imaging

After >2 weeks recovery from surgery, GCaMP6s fluorescence was imaged using a Bruker Investigator two-photon microscopy system with a resonant galvo-scanning module. Posterior dysgranular RSC was targeted by imaging <1 mm lateral of the sagittal sinus and <1 mm anterior to the transverse sinus.

For fluorescence excitation, we used a Ti:Sapphire laser (Mai-Tai eHP, Newport) with dispersion compensation (DeepSee, Newport) tuned to $\lambda = 920$ nm. For collection, we used GaAsP photomultiplier tubes (Hamamatsu). To achieve a wide field of view, we used a 16× 0.8 NA microscope objective (Nikon) and imaged at a $414 \times 414$ μm, $690 \times 690$ μm, or $829 \times 829$ μm field of view spanning $760 \times 760$ pixels. Laser power ranged from 40 to 75 mW at the sample depending on GCaMP6s expression levels. Photobleaching was minimal (<1% min$^{-1}$) for all laser powers used. A custom stainless-steel light blocker (eMachineShop) was mounted to the head plate and interlocked with a tube around the objective to prevent ambient light from reaching the PMTs.

## Two-photon post-processing

Images were acquired using PrairieView acquisition software (Bruker) at 10 Hz and converted into TIF files. All subsequent analyses were performed in MATLAB (Mathworks) using custom code (https://github.com/ucsb-goard-lab/Two-photon-calcium-post-processing). First, images were corrected for X-Y movement by registration to a reference image (the pixel-wise mean of all frames) using 2-dimensional cross correlation.

To identify responsive neural somata, a pixel-wise activity map was calculated using a modified kurtosis measure. Neuron cell bodies were identified using local adaptive threshold and iterative segmentation. Automatically defined ROIs were then manually checked for proper segmentation in a graphical user interface (allowing comparison to raw fluorescence and activity map images). To ensure that the

response of individual neurons was not due to local neuropil contamination of somatic signals, a corrected fluorescence measure was estimated according to:

$$F_{\text{corrected}}(n) = F_{\text{soma}}(n) - \alpha\left(F_{\text{neuropil}}(n) - \bar{F}_{\text{neuropil}}\right) \quad (1)$$

where $F_{\text{neuropil}}$ was defined as the fluorescence in the region <30 μm from the ROI border (excluding other ROIs) for frame $n$. $\bar{F}_{\text{neuropil}}$ is $F_{\text{neuropil}}$ averaged over all frames. $\alpha$ was chosen from [0 1] to minimize the Pearson's correlation coefficient between $F_{\text{corrected}}$ and $F_{\text{neuropil}}$. The $\Delta F/F$ for each neuron was then calculated as:

$$\Delta F/F = (F_n - F_0)/F_0 \quad (2)$$

where $F_n$ is the corrected fluorescence ($F_{\text{corrected}}$) for frame $n$ and $F_0$ defined as the mode of the corrected fluorescence density distribution across the entire time series.

**De-rotation of image time series.** Because the microscope is fixed relative to the mouse in the head rotation experiments, the images from the microscope were de-rotated in order to be properly processed[66]. The initial non-rotation period was used to create a template that the remainder of the images would be registered to. Registration was performed by maximizing the Mattes mutual information between each frame and the template via a one plus one optimizer (MATLAB, Mathworks). The geometric transformation of the previous successfully registered image was applied as an initial transform in order to optimize the performance and speed of the registration. After de-rotating each frame, an occupancy map was calculated from the resulting images, and the image was cropped so that only areas that contained all frames were included in further analyses. The final image series was subsequently analyzed using the two-photon processing pipeline described above. All the associated code for derotation can be found here: https://github.com/ucsb-goard-lab/2P-Derotation.

**Axon terminal imaging.** For axon terminal imaging, we processed the TIF files using the Python implementation of Suite2P[83]. Briefly, TIFs underwent rigid registration using regularized phase correlations. Regions of interest were extracted via clustering correlated pixels, and were manually checked based on location, morphology, and $\Delta F/F$ activity. A specific configuration of Suite2P was used that is optimized for detecting axon processes[84]. After defining ROIs, we used custom code (MATLAB) to check that the same axon process was not sampled multiple times (for ROIs with Pearson correlation >0.5, all ROIs except one were excluded).

## Analysis of calcium data

To avoid low pass filtering of heading tuning curves due to slow calcium dynamics, we used a MATLAB implementation of a sparse, non-negative convolution algorithm (OASIS) on $\Delta F/F$ traces[85] with an autoregressive model of order 1 for the convolution kernel.

For the floating chamber experiments with voluntary control, the Rayleigh vector length (RVL) was used to determine if a cell was heading selective. The RVL was calculated for each cell and compared against a shuffled distribution. To create the shuffled distribution, each cell's spike data was circularly shuffled and the tuning curve and resulting RVL was calculated. This was repeated 1000 times to create a distribution of shuffled RVLs. Cells whose true RVL met or exceeded the 99th percentile of this distribution were considered to be heading selective.

For calculation of heading selectivity in the rotating chamber experiments, each cell's spike time series was divided into trials, with each trial representing a single full rotation of the chamber. To calculate tuning curves, the orientation of the chamber at each frame was first binned into 60 bins of 6 degrees. Then, the response was

calculated for each trial using the spike rate of the cell at each heading bin, smoothed by a 15° moving average filter. The overall tuning curve was calculated by averaging the responses across all trials.

Next, we determined whether each cell (or axon terminal) was significantly heading selective. First, the reliability each cell's heading preference across trials was calculated by randomly splitting the trials into two groups and calculating the correlation coefficient between the resulting tuning curves. Then, each trial's activity was circularly shifted prior to calculating the correlation coefficient. This was repeated 1000 times for each cell, creating a real and shuffled distribution, which was compared via a two-sample Kolmogorov–Smirnov test.

Cohen's $d$ was calculated to gauge the separation between the two distributions. A cell was considered reliable if it passed the two-sample Kolmogorov–Smirnov test at $p < 0.1$ and had a Cohen's $d > 0.8$. In addition, A single-term (for single cue) or two-term (for dual cue) Gaussian was fit to the trial averaged tuning curve as well as the trial averaged shuffled tuning curves. As before, the tuning curves were shuffled and fit 1000 times. The goodness of fit ($r^2$) of the true fit was compared against the 90th percentile of $r^2$ for the shuffled fits to determine if the cell's tuning curve had the proper shape. These threshold values were chosen so that the combined significance value of the two measures is $p = 0.01$. When repeating these procedures using pre-shuffled data, no cells passed these criteria, suggesting that the number of heading selective cells is accurate, and not due to chance.

To compare tuning curves across cells regardless of preferred direction, a cross-validated alignment was performed. For each trial, the responses from every other trial were used to construct a tuning curve, and the peak index of that tuning curve was used to align the held out trial. For light-on and light-off experiments, individual tuning curves for each condition were first calculated by only averaging trials that belonged to each light condition. Cross-validated alignment was performed on the light-on tuning curves, and the offsets applied to the light-off tuning curves, so that the same circular offset was applied to both conditions.

**Calculation of coherent remapping.** To determine whether or not a population of neurons remapped coherently, we created cross-validated procedure for testing individual cells. For each neuron in a recording, the difference between its preferred direction in the first and second context was calculated. Next, all other cells in the recording were used to calculate an "expected offset" by taking the median across all cell pair differences in preferred direction across contexts. Lastly, the difference between the neuron of interest and the median difference was taken to determine if the neuron of interest remapped coherently with the remainder of the population. This procedure can be summarized as follows:

$$\Delta \text{phase offset}_n = \left( \text{pref}_{(A,n)} - \text{pref}_{(B,n)} \right) - \left( \widetilde{\text{pref}}_{(A,[0,N] \neq n)} - \widetilde{\text{pref}}_{(B,[0,N \neq n)} \right) \tag{3}$$

where $\text{pref}_A$ and $\text{pref}_B$ is the preferred direction in context A or B, respectively, of neuron $n$ and $N$ is the total number of neurons. $\widetilde{\text{pref}}$ denotes the median of preferred directions. Perfect remapping would result in a difference of 0, suggesting that each neuron exhibited the exact same heading offset as the rest of the population. This process was repeated for each recording to measure the amount of coherent remapping for each neuron in each recording. These values were compared against a shuffled distribution, in which random neurons were chosen and compared, rather than defined cell pairs.

**Clustering of RSC soma responses.** To categorize the response profiles of RSC somata, unsupervised clustering was performed. First, all cells were fit with a sum of Gaussians as defined below for both light-

on and light-off conditions:

$$f(x) = a_0 + a_1 \times e^{-((x-b_1)/c_1)2} + a_2 \times e^{-((x-b_2)/c_2)2} \tag{4}$$

where $a_0$ is the baseline offset, $a_1$, $b_1$, $c_1$, $a_2$, $b_2$, and $c_2$ are the amplitude, location, and standard deviation for the first and second peaks, respectively. The coefficients $a_1$, $a_2$, $c_1$, and $c_2$ for each the light-on and light-off tuning curves were fit with an 8 component Gaussian mixture model (GMM). Coefficients $b_1$ and $b_2$ were omitted so that the location of the peak was not a determining factor in clustering. A cluster evaluation using silhouette analysis was performed to determine the optimal number of clusters, which reported a local maximum at $k = 3$ clusters, resulting in a 3 component GMM. Each neuron's Mahalanobis distance was calculated from each centroid, and the minimum distance among centroids was taken. Then, any outliers whose minimum distance was significantly greater was assigned to a null cluster. In order to separate the neurons into specific clusters, the outputs of the GMM were used to create archetypal tuning curves that represented these clusters in light-on and light-off conditions. The cosine distance of each neuron's light-on and light-off tuning curve was calculated against each archetypal tuning curve, and the summed minimum distance was used to determine the best fitting cluster for each cell.

Because of the high dimensionality of the GMM components, a dimensionality reduction was performed for visualization (Fig. 5B). A linear discriminant analysis model (LDA) was trained on the outputs of the GMM, and the weights from the trained model were used to reduce the dimensionality of the real data to the first two dimensions for visualization.

**Clustering of axonal responses.** To project the visual and ADN axonal responses on the same axes as the somatic data, the trained LDA was used to reduce the dimensionality of the axonal data. Bounding ellipses were drawn based on the calculated 95% CI of the points in each cluster.

**Calculation of bimodality.** To represent the bimodality of a neuron, we used a flip score metric also described previously[46]. Briefly, an autocorrelation of each cell's tuning curve was performed. A bimodal cell resulted in a peak centered at 180°, which was absent in unimodal cells. The flip score was then calculated as follows:

$$\text{Flip score} = CC_{180°} - (CC_{90°} + CC_{270°})/2 \tag{5}$$

where $CC_{180°}$, $CC_{90°}$, and $CC_{270°}$ are the correlation coefficients of the autocorrelation at the 180°, 90°, and 270° positions, respectively.

## Decoding RSC somatic activity

**Preprocessing and alignment of calcium data.** In order to aggregate trial data across all recordings, the following preprocessing steps were performed. First, each rest period in which the chamber does not rotate between light-on and light-off conditions was identified. Individual trials were then defined as full rotations either preceding or following the rest period, so that each trial contained a full rotation of the chamber. For light-off to light-on conditions, the rest period was identified that separated light-off from light-on periods. By defining full trials about the rest period, each trial was ensured to fully span 360°. Next, minor variations in the speed of rotation between individual trials were accounted for by resampling both the heading information and neural data to be equal length (200 bins) across all trials in all recordings.

**Decoding strategy.** After preprocessing, the data were aggregated and the raw calcium data were decoding using Bayesian decoding[86]. Data were split into even and odd timepoints for training and testing. All reported results are with testing data only.

**Calculating decoder performance metrics.** Because each pseudo-population had uneven numbers of neurons, we first performed a bootstrapping procedure prior to calculating performance metrics. We sampled, with replacement, neurons from each group, with the number of neurons determined by the size of the smallest group ($n = 332$ cells). For the "all tuned cells" group, we sample randomly with replacement from all three cell classes, preserving the rough proportions of each response class. The analysis was performed on sampled subpopulations for 1000 iterations to calculate the mean and bootstrapped s.e.m. at each time point. The decoder error for each frame was calculated as follows:

$$\text{Decoder error} = |H_{\text{decoded}} - H_{\text{actual}}| \qquad (6)$$

where $H_{\text{decoded}}$ is the decoded heading at each frame and $H_{\text{actual}}$ is the actual heading at each frame. Because of the overrepresentation of 180° errors due to the landmark cell contribution, we found that the average decoder error did not well represent the actual trial performance. Therefore, in addition to decoder error, we calculated the decoder accuracy for each trial as follows:

$$\text{Decoder accuracy} = \sum(\text{Decoder error} < 18°)/N_{\text{frames}} \times 1000 \qquad (7)$$

where $N_{\text{frames}}$ is the total number of frames in the current trial.

Decoder accuracy was displayed as bootstrapped plots with error bars denoting the 2.5th and 97.5th percentile. If the confidence interval did not overlap chance (0.18), the decoder accuracy was deemed to be statistically significant. The specific decoding and decoder error examples in the Fig. 6 plots were taken from the iteration with median performance based on decoder accuracy to illustrate a representative decoding session.

## Statistical information

To test decoding error across recordings, paired t-tests were performed. To test whether angular data were clustered around a mean angle, a V-test was performed (Circular Statistics Toolbox, MATLAB). To compare the differences between two distributions, a two-sample Kolmogorov–Smirnov test was performed. To compare differences across multiple animals, generalized linear mixed-effects models were used to control for correlations of cells within the same animal. For bootstrapped samples, the p-values were calculated as one minus the fraction of iterations that passed a Student's t-test or Kolmogorov–Smirnov test at an alpha ≤0.05. All confidence intervals were calculated from bootstrapped data. When performing bootstrapping, data were randomly sampled with replacement. Statistical comparisons between groups were performed as two-tailed tests.

## Reporting summary

Further information on research design is available in the Nature Portfolio Reporting Summary linked to this article.

# Data availability

The data generated in this study have been deposited on the following Dryad repository: https://doi.org/10.25349/D91G8Q. Source data are provided with this paper.

# Code availability

All of the code for generating Figs. 1–6 is available on the following GitHub repository: https://github.com/ucsb-goard-lab/Neurotar-HD-Experiments.

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

## Acknowledgements

We would like to thank Pierre-Yves Jacob, Kate Jeffery, Sung Soo Kim, Patrick LaChance, Cris Niell, Jeffrey Taube, and Ningyu Zhang for the discussion of the manuscript. This work was supported by grants to M.J.G. from NIH (R01 NS121919), NSF (1934288), and the Whitehall Foundation.

## Author contributions

K.K.S. and M.J.G. designed the experiments and analyses. K.K.S. conducted the experiments and analyzed the data. K.K.S. and M.J.G. wrote the paper.

## Competing interests

The authors declare no competing interests.
