## [Peer Review File · Nature Communications]

Coregistration of heading to visual cues in retrosplenial cortexREVIEWER COMMENTS

Reviewer #1 (Remarks to the Author):

Sit and Goard characterize calcium activation of dysgranular retrosplenial cortex (RSC) neurons in head-fixed mice as a function of perceived orientation in the environment. In a series of manipulations the authors demonstrate that RSC “heading” neurons (HD) recorded under this experimental preparation exhibit similar properties as those observed in freely moving animals, namely that tuning preferences span the full angular range and that the network exhibits coherent remapping across visual contexts. By imaging axon terminals in RSC arising from visual areas (VIS) and the anterodorsal thalamus (AD) during light and dark manipulations, Sit and Goard show that HD information arises from both regions in a context dependent manner hinged on visual input. The authors categorize heading, landmark, and alignment cells which are reliant on AD, visual, and their integration, respectively. The manuscript is written and presented clearly and I have very few technical comments overall. While I believe the findings will be of interest to a broad community of scientists interested in spatial processing and navigation, I have a few comments that I believe should be addressed in a revised manuscript.

It was unclear to me how alignment cells add to or differ from prior observations of bimodal and landmark modulated HD cells in the same or neighboring brain regions (Jacob et al., 2017; LaChance et al., 2022). The authors address this in the discussion but I was unfortunately left unsatisfied by the distinctions presented. I think this partly arises because the term “alignment” seems defined arbitrarily despite this population resembling these prior reports. It is unclear what “alignment” these neurons actually perform and what evidence of “alignment” is presented (although I appreciate the hypothetical models proposed). For instance, if alignment cells are truly responsible for registering visual and heading orienting signals, why do they not exhibit distinct dynamics at the transition between light and dark (they seem similar to the heading population in nearly every metric)? Co-activation analyses or a deeper look into the temporal resolution of the emergence of HD tuning at light/dark transitions could reveal a unique influence of alignment cells over heading or landmark populations.

Given the lack of distinct clusters in the dimensionality reduced space it seems entirely plausible that the alignment cell population is simply a noisier version of either landmark or heading cells. A robust analysis of tuning reliability across trials for each tuning peak should be conducted between groups. The authors should look at other metrics/methods for clustering these cells that may yield more distinct clusters. As it stands, it is unclear if these neurons should truly be distinguished as a distinct population.

Relatedly, a key implication of the alignment population is that they arise from intra-RSC integration dynamics. However, the authors do not look for alignment-like responses from V1 and AD terminals. This should be investigated using the same classification algorithms as the somata.

In the discussion the authors indicate that a key advance from this work is the discovery that heading responses can be observed in head-fixed mice with no vestibular input. In my mind this has already been shown in Vogits et al. 2017 and Heneestad et al., 2021 (although in these works this observation was perhaps not as explicitly stated). Something interesting that the authors could potentially add to prior work is an examination of whether the head-fixed heading cells exhibit the same tuning properties in the free floating versus fixed rotation conditions. Maybe I missed it but do the authors examine this? It appears the primary comparison is fixed rotation of the animal or fixed rotation of the environment with the animal still.

Given the experimental configuration I wonder if heading responses can be decoupled from egocentric bearing to a visual cue? Do the authors analyze tuning curves during both CW and CCW rotations of the animal (this wasn't clear from the methods)? In the free floating condition it would also be interesting to look at preferred heading as a function of proximity from the center.

L91 - More of a comment - the deconvolution method is appropriate but I noticed in the methods that the HD tuning curves are smoothed. If memory serves, smoothing of HD tuning curves is not commonly done and is a bit counter to claims regarding excessive smoothing.

L559-563 - "Following clustering by the GMM, an additional exclusion step was performed to remove cells with poor tuning curves..." This seems a bit circular. What do the average cluster responses look like without this step?

L378 - Axon tracing experiments?

L392 - Are there differences in terminal responses between AM and PM? I assume these animals are grouped into the VIS category?

L582 - As I understand it, either AD OR VIS terminals were imaged in an individual animal. Accordingly, this metric of input influence is based on the similarity between individual somas in one animal and terminal population activation patterns across all animals? Is this correct? If so, is the input index different for within versus across animal comparisons (i.e. an AD animal to within and across AD versus the same AD animal to VIS animals)?

Evidence of the presence of all main findings in all animals should be presented wherever possible.

Reviewer #2 (Remarks to the Author):

The manuscript by Sit & Goard addresses how a code for head direction can be aligned to visual landmarks by studying large scale neuronal activity in dysgranular retrosplenial cortex of mice. The authors employ a floating platform that mice can freely move on or respond accordingly to experimenter-controlled rotations. By investigating GCaMP responses of RSC cells to the relative orientation of a visual cue on the wall, the authors identify a subset of cells with reliable “heading” responses. By matching the same neurons under platform rotation vs head-rotation conditions, the authors conclude that these heading cells have the same properties as head direction cells found in the freely rotating rodents. However, these cells do seem not require vestibular input. The authors identify three groups of neuronal responses in RSC based on unique tuning to light-on and -off conditions in a visually symmetric environment. The authors further link average responses to those described for two major inputs to RSC (ADN carrying HD versus AM/PM areas carrying visual inputs) through axonal imaging. Finally, the authors report different contributions to heading representations of each subgroup through heading decoding performance.

The paper is well written, the experimental set up to tease apart different responses in RSC is clever, and the findings on circuit mechanisms for landmark alignment are broadly exciting for the field. The work is generally well conducted, impactful, and important, representing progress in the challenging field of understanding how RSC works and how landmarks are integrated into head direction processing. These are key unknowns in the larger picture of understanding the neural basis of complex navigation in mammals. Some additional experiments and editing would strengthen the claims made in the manuscript and would be useful for clarifying the different roles for the 3 classes of cells described

Major comments:

1) It would be helpful to see whether the tuning preferences observed in Fig. 2 are maintained under purely visual stimuli, namely as the walls of the arena are rotated and the visual cues are passively experienced. The authors emphasize the importance of proprioceptive signals for heading responses, however Shinder et al, 2011 have shown that motor efference copy and proprioceptive signals are not necessary for head direction coding in areas other than RSC. In Fig. 5 the authors show that a large proportion of tuned cells in RSC are actually visual (or landmark) cells. Would the authors expect that under exclusive visual rotation conditions we would also see these kinds of responses? Along those lines, Fig. 2b is limited to the matched heading responses between the two conditions. It is however unclear if there are heading (or head direction) cells that emerge or are silenced between the two conditions. A more thorough characterization of the activity under the two conditions would help provide more context and a more concrete interpretation regarding the role of RSC in heading.

2) In Fig. 5E, it was not clear how the authors explain the second bump of the alignment cells during the light off. It seems similarly attenuated in the light-off condition as the preferred bump, even though the argument in the text is that the alignment cells respond differently in the two conditions. Is it possible this is a result of the task design, with the platform rotations occurring during light off as well? Additional experiments changing the rate or direction of rotations to eliminate prediction responses would strengthen the argument.

3) It is unclear if the authors are stating that the alignment only group have significantly improved decoding performance than the heading only group. Even though the authors caution that there might be a continuum in the population, the argued differences in the performance suggest different roles, as illustrated in Fig. 6f. Clarity on this point would be useful in evaluating the decoding results.

4) The authors indicate that the decoding performance of the heading only cluster is lower in the light off condition than in light on based on the 95% CI, but the tuning and width of the heading cluster in figure 5c seem preserved and comparable between the two conditions, suggesting that the drift in darkness observed in previous studies in the freely moving rodents is not really playing a role in this case. Can the authors speculate as to why the decoding might then be worse?

5) Fig. 6b-e show an apparent drop in decoder accuracy prior to lights being turned off or persistence after they are turned on. It is unclear how such drops can be explained. This may hint at a flaw or uncontrolled variable in either experimental design or data analysis. A clear explanation of this point seems necessary to understand what this effect is.

6) In line 283-284, it is unclear if light-on and off conditions are pooled for calculating decoding accuracy difference for the mixed population vs all subgroups. The authors should address this point.

Minor Comments:

1) Line 63 and in the references Zhang et al should be 2021 not 0009;

2) Line 168, I would specify "for eliciting heading responses in RSC if vision and proprioception are spared".

3) About half of the ADN responses in figure 4D do not seem clearly unimodal. It is possible this variability originates from the head-fixation, however a histology confirmation of the injection target would provide more confidence.

4) The authors use the word "landmark" but given the symmetric arrangement and the nature of the behavior, the authors should use "visual cue-driven" or "visual" cells.

5) Line 221 it seems only 13% of the total cells are tuned – are these the "all RSC cells" shown in Fig 5A?

- 6) Line 328 I would change heading to HD – the past literature mostly focused on the freely moving, vestibular- driven HD.
- 7) Line 445 typo, “independent”
- 8) Line 459 the sampling frequency is not mentioned.
- 9) Line 565 should be figure 5B.
- 10) Line 571 and later 577 the authors should cite the previous work they refer to.
- 11) Line 571 it is unclear what gls in the glsrvl acronym stands for
- 12) First part of equation 7 should have a minus sign.
- 13) Comparisons are provided for panels 5G and 5H, but not for 5I
- 14) Supplemental Fig3 example F has no detectable rates for the 180 degree, even though a small preference does appear in the polar plot on the right.
- 15) Is the pseudo-population decoding based on equal samples of the three populations or does it reflect the proportions in figure 5F (including the unclassified cells)?

REVIEWER COMMENTS

We thank both reviewers for their time and effort in reading our manuscript and providing valuable feedback to help strengthen our manuscript. We have revised the manuscript in order to respond to specific reviewers comments below. However, we also made some additional changes to our visualization and analysis in order to strengthen the findings. First, the majority of the plots showing the heading tuning plots have been changed from Cartesian to polar plots in order to be more consistent with previous work. We have kept Cartesian plots for specific figures that benefit from that type of presentation, such as when showing changes in preferred direction of tuning curves. Second, we have also updated the decoding strategy to better capture the contributions of each cell class in the RSC, and have updated the figure and text accordingly.

Reviewer #1 (Remarks to the Author):

Sit and Goard characterize calcium activation of dysgranular retrosplenial cortex (RSC) neurons in head-fixed mice as a function of perceived orientation in the environment. In a series of manipulations the authors demonstrate that RSC “heading” neurons (HD) recorded under this experimental preparation exhibit similar properties as those observed in freely moving animals, namely that tuning preferences span the full angular range and that the network exhibits coherent remapping across visual contexts. By imaging axon terminals in RSC arising from visual areas (VIS) and the anterodorsal thalamus (AD) during light and dark manipulations, Sit and Goard show that HD information arises from both regions in a context dependent manner hinged on visual input. The authors categorize heading, landmark, and alignment cells which are reliant on AD, visual, and their integration, respectively. The manuscript is written and presented clearly and I have very few technical comments overall.

While I believe the findings will be of interest to a broad community of scientists interested in spatial processing and navigation, I have a few comments that I believe should be addressed in a revised manuscript.

It was unclear to me how alignment cells add to or differ from prior observations of bimodal and landmark modulated HD cells in the same or neighboring brain regions (Jacob et al., 2017; LaChance et al., 2022).

It is likely that the cells we observe in our study overlap with previously observed bimodal cells in RSC and other regions. However, in our study, we experimentally dissociate the visual and heading contributions. This reveals cell types that would likely appear identical in previous studies. For example, both landmark and alignment cells exhibit bidirectional responses in symmetrical environments, but they have very different responses when visual input is removed. As a result, although this work is certainly complementary to Jacob et al. 2017 and LaChance et al. 2022, as well as related modeling work (Yan 2021), it reveals novel response types and suggests how visual and heading responses might be integrated. We have addressed this in lines 338-343 in the Discussion.

The authors address this in the discussion but I was unfortunately left unsatisfied by the distinctions presented. I think this partly arises because the term “alignment” seems defined arbitrarily despite this population resembling these prior reports. It is unclear what “alignment” these neurons actually perform and what evidence of “alignment” is presented (although I appreciate the hypothetical models proposed). For instance, if alignment cells are truly responsible for registering visual and heading orienting signals, why do they not exhibit distinct dynamics at the transition between light and dark (they seem similar to the heading population in nearly every metric)? Co-activation analyses or a deeper look into the temporal resolution of the emergence of HD tuning at light/dark transitions could reveal a unique influence of alignment cells over heading or landmark populations.

The reason we refer to them as “alignment” cells is that the heading-like response (unimodal response in dark) is always co-tuned with one of the two peaks of the landmark-like response in light. As a result, there is good reason to think these cells play a role in co-registering these two signals. We agree that proof of this hypothesis will likely require targeted perturbations of distinct cell types, which is beyond the scope of the current study.

On the suggestion of the reviewer, we have improved the decoder analysis to better capture the coding capability of each response type. Although single time point decoding is still too variable to illuminate the immediate effects of light/dark transitions, the improved analysis reveals that the three cell classes exhibit significantly different decoding performance in light and dark (Figure 6B, right).

Given the lack of distinct clusters in the dimensionality reduced space it seems entirely plausible that the alignment cell population is simply a noisier version of either landmark or heading cells. A robust analysis of tuning reliability across trials for each tuning peak should be conducted between groups.

The authors should look at other metrics/methods for clustering these cells that may yield more distinct clusters. As it stands, it is unclear if these neurons should truly be distinguished as a distinct population.

All the neurons represented in the clustering have already passed an initial reliability measurement to ensure that the peaks are reliable across trials. We have tried several unsupervised centroid-based and density based clustering approaches, which generally agree that 3 clusters are appropriate for our data in the full dimensional space (Lines 560 - 562). The apparent overlap of the clusters is partially due to the reduced dimensionality we used for visual representation, as there is reasonably good cluster separation in the full 8D space. That said, as we are clear to acknowledge in the manuscript, the responses do lie along a continuum. Although cells that clearly belong to one response type, others are more difficult to distinguish. We have provided additional examples of RSC responses in Figure 5C-E and Figure S4C-E to clarify this point.

Relatedly, a key implication of the alignment population is that they arise from intra-RSC integration dynamics. However, the authors do not look for alignment-like responses from V1

and AD terminals. This should be investigated using the same classification algorithms as the somata.

Using the same classification steps as for the RSC somata show that the population of visual terminals and ADN terminals are relatively homogenous, without any clear cluster separation reported by the cluster evaluation. In contrast, applying the same cluster evaluation on RSC somatic responses suggests the existence of 3 clusters. To investigate further, we plotted the responses of ADN and VIS terminals using the same linear discriminant axes we used for the RSC clustering (Figure 5I). Although there is some variability, we find that ADN terminals are almost entirely restricted to the heading cluster and that VIS terminals are almost entirely restricted to the landmark cluster.

In the discussion the authors indicate that a key advance from this work is the discovery that heading responses can be observed in head-fixed mice with no vestibular input. In my mind this has already been shown in Vogits et al. 2017 and Heneestad et al., 2021 (although in these works this observation was perhaps not as explicitly stated).

In both Voigts 2017 and Heneestad 2021, there was still physical rotation of the animal's head, and thus vestibular drive. Our experiments are the first to show heading direction responses in the complete absence of vestibular modulation. We have updated the discussion to make this point more clear (line 307-309).

Something interesting that the authors could potentially add to prior work is an examination of whether the head-fixed heading cells exhibit the same tuning properties in the free floating versus fixed rotation conditions. Maybe I missed it but do the authors examine this? It appears the primary comparison is fixed rotation of the animal or fixed rotation of the environment with the animal still.

We added new experiments with paired recordings measuring responses of the same cells across free floating and fixed rotation conditions and found that neurons generally preserve their tuning across these conditions. We added these results to a new supplemental figure (Figure S6).

Given the experimental configuration I wonder if heading responses can be decoupled from egocentric bearing to a visual cue? Do the authors analyze tuning curves during both CW and CCW rotations of the animal (this wasn't clear from the methods)?

We added recordings that have interleaved CW and CCW trials and do not see any significant difference in the tuning curves of neurons between these conditions. We have added the results to a new supplemental figure (Figure S5).

In the free floating condition it would also be interesting to look at preferred heading as a function of proximity from the center.

We have tried this analysis and found no significant difference in tuning curves as a function of proximity from the center of the cage (Figure R1). First, we divided the arena into near and far zones (Figure R1A). Tuning curves were then independently calculated, using only data from when the mouse was in that zone (Figure R1B). Lastly, the preferred direction of these neurons was calculated and compared across zones, showing that the neurons' preferred direction stays highly stable across zones (Figure R1C). Unfortunately, we found that subdividing the chamber into a greater number of zones decreased the quality of our tuning curves due to a paucity of sampling. We have included the 2-zone analysis as a reviewer figure.

Reviewer Figure 1. Heading direction tuning does not depend on location within the floating chamber. **A)** The chamber was divided into two concentric zones and the heading direction tuning for each cell was measured independently in each zone. **B)** Two example cells showing overlapping heading direction tuning in the near distance (orange) and far distance (blue) zones. **C)** Histogram of difference in preferred direction between near distance and far distance zones within cells (orange) versus between shuffled cells (gray).

L91 - More of a comment - the deconvolution method is appropriate but I noticed in the methods that the HD tuning curves are smoothed. If memory serves, smoothing of HD tuning curves is not commonly done and is a bit counter to claims regarding excessive smoothing.

Previous papers with electrophysiological recordings (e.g., see Giocomo 2014, etc) used a similar method as ours, with a 15 degree smoothing filter. Without any smoothing, the curves become very jagged due to the limited number of frames per direction.

L559-563 - "Following clustering by the GMM, an additional exclusion step was performed to remove cells with poor tuning curves..." This seems a bit circular. What do the average cluster responses look like without this step?

We agree, though we found that adding a second step helped remove neurons that did not match their assigned cluster. This step has been revised: rather than using the tuning curves for exclusion, we looked at the minimum Mahalanobis distance from each data point to the centroid of each Gaussian. We then removed points outside this threshold to avoid disrupting average cluster responses, defined as data points greater than a defined threshold from any Gaussian centroid (lines 562-568 in Methods section).

L378 - Axon tracing experiments?

Thank you, this has been corrected.

L392 - Are there differences in terminal responses between AM and PM? I assume these animals are grouped into the VIS category?

For experiments involving virus injection into higher visual cortex, we injected into both areas AM and PM, so any differences between responses from the terminals could not be parsed from our data.

L582 - As I understand it, either AD OR VIS terminals were imaged in an individual animal. Accordingly, this metric of input influence is based on the similarity between individual somas in one animal and terminal population activation patterns across all animals? Is this correct? If so, is the input index different for within versus across animal comparisons (i.e. an AD animal to within and across AD versus the same AD animal to VIS animals)?

We decided to replace this analysis with another one that better illustrates our point. The new analysis is shown in Figure 5I.

Evidence of the presence of all main findings in all animals should be presented wherever possible.

We have changed the reporting of the fraction of each cell to be the mean and standard error across recordings rather than the percentage of all pooled cells to show that each recording contains these cell types (Lines 217-219).

Reviewer #2 (Remarks to the Author):

The manuscript by Sit & Goard addresses how a code for head direction can be aligned to visual landmarks by studying large scale neuronal activity in dysgranular retrosplenial cortex of mice. The authors employ a floating platform that mice can freely move on or respond accordingly to experimenter-controlled rotations. By investigating GCaMP responses of RSC cells to the relative orientation of a visual cue on the wall, the authors identify a subset of cells with reliable "heading" responses. By matching the same neurons under platform rotation vs head-rotation conditions, the authors conclude that these heading cells have the same properties as head direction cells found in the freely rotating rodents. However, these cells do seem not require vestibular input. The authors identify three groups of neuronal responses in RSC based on unique tuning to light-on and -off conditions in a visually symmetric environment. The authors further link average responses to those described for two major inputs to RSC (ADN carrying HD versus AM/PM areas carrying visual inputs) through axonal imaging. Finally, the authors report different contributions to heading representations of each subgroup through heading decoding performance.

The paper is well written, the experimental set up to tease apart different responses in RSC is clever, and the findings on circuit mechanisms for landmark alignment are broadly exciting for the field. The work is generally well conducted, impactful, and important, representing progress in the challenging field of understanding how RSC works and how landmarks are integrated into head direction processing. These are key unknowns in the larger picture of understanding the neural basis of complex navigation in mammals. Some additional experiments and editing would strengthen the claims made in the manuscript and would be useful for clarifying the different roles for the 3 classes of cells described

Major comments:

1) It would be helpful to see whether the tuning preferences observed in Fig. 2 are maintained under purely visual stimuli, namely as the walls of the arena are rotated and the visual cues are passively experienced.

We have added a set of experiments that compare responses from the same cells for our standard rotation condition versus wall rotation only. We find that each cell class is differentially affected by this manipulation. Briefly, if we consider a “heading component” and a “visual component” to each cell class, we can view the landmark cluster as being purely driven by the visual component, the heading cluster being purely driven by the heading component, and the alignment cell as having a mix of the two. By removing other sensory signals, but sparing visual cues, we show that the heading components are strongly reduced while the visual components are largely unaffected. We have included the results of these experiments in Figure 5J.

The authors emphasize the importance of proprioceptive signals for heading responses, however Shinder et al, 2011 have shown that motor efference copy and proprioceptive signals are not necessary for head direction coding in areas other than RSC.

Although we agree that motor efference copy and proprioceptive signals are not necessary for the generation of heading responses, they have important roles in updating and maintaining heading responses. Indeed removal of these motor and proprioceptive signals affect the responses of HD cells, even in the ADN (Taube 1995, Knierim et al 1995). Therefore, we believe that in the absence of vestibular modulation, as in our set-up, the information from motor and proprioceptive sources may be sufficient for driving head direction coding in the head direction network. This is supported by the results of Figure 5J-K and supports a view that multiple sensory signals are integrated in updating of the heading direction signal (see Discussion lines 328-343).

In Fig. 5 the authors show that a large proportion of tuned cells in RSC are actually visual (or landmark) cells. Would the authors expect that under exclusive visual rotation conditions we would also see these kinds of responses?

As discussed in response to the previous comment, we have addressed this with new experiments comparing full chamber rotation to visual cue rotation (Figure 5J-K).

Along those lines, Fig. 2b is limited to the matched heading responses between the two conditions. It is however unclear if there are heading (or head direction) cells that emerge or are silenced between the two conditions. A more thorough characterization of the activity under the two conditions would help provide more context and a more concrete interpretation regarding the role of RSC in heading.

It is difficult to quantify the number of appearing or disappearing neurons between the two conditions because many of the cells are lost in the rotating field experiments due to the preparation (e.g., cells near the periphery often exhibit signal loss due to small z-plane movements or moving out of the imaged field). As a result, we focused our analyses solely on the cells that were clearly present and tuned across both recordings.

2) In Fig. 5E, it was not clear how the authors explain the second bump of the alignment cells during the light off. It seems similarly attenuated in the light-off condition as the preferred bump, even though the argument in the text is that the alignment cells respond differently in the two conditions. Is it possible this is a result of the task design, with the platform rotations occurring during light off as well? Additional experiments changing the rate or direction of rotations to eliminate prediction responses would strengthen the argument.

The difference in the alignment cell response is mainly that the tuning changes from a bimodal curve to a unimodal curve - the average responses appear more bimodal because the dark peak was sometimes aligned to the smaller light on peak. We have revised the figure to better illustrate this by comparing flip scores in the light-on and light-off condition for each cluster.

3) It is unclear if the authors are stating that the alignment only group have significantly improved decoding performance than the heading only group. Even though the authors caution that there might be a continuum in the population, the argued differences in the performance suggest different roles, as illustrated in Fig. 6f. Clarity on this point would be useful in evaluating the decoding results.

To help address this point, we have changed the decoding strategy, increasing overall performance and better revealing differences in decoding performance across cell classes.

4) The authors indicate that the decoding performance of the heading only cluster is lower in the light off condition than in light on based on the 95% CI, but the tuning and width of the heading cluster in figure 5c seem preserved and comparable between the two conditions, suggesting that the drift in darkness observed in previous studies in the freely moving rodents is not really playing a role in this case. Can the authors speculate as to why the decoding might then be worse?

With our updated decoder analysis, the main decrease we see in decoding performance is in the first two trials following the switch to dark. We suspect that longer dark periods would result in more gradual drift, but our dark periods did not extend that far. We suspect the decreased performance is due to the lower gain and decreased reliability seen in heading and alignment cells in the dark.

5) Fig. 6b-e show an apparent drop in decoder accuracy prior to lights being turned off or persistence after they are turned on. It is unclear how such drops can be explained. This may hint at a flaw or uncontrolled variable in either experimental design or data analysis. A clear explanation of this point seems necessary to understand what this effect is.

Our apologies, the apparent changes were due to a bug in the code that sometimes misattributed parts of trials to the wrong side of the light/dark transition. This has been fixed in the new version of the decoder.

6) In line 283-284, it is unclear if light-on and off conditions are pooled for calculating decoding accuracy difference for the mixed population vs all subgroups. The authors should address this point.

This statement has been removed from the text because the newest decoder analyses do not support this point. Rather, using all cell groups, the decoder takes advantage of each group to perform maximally in both light-on and light-off conditions.

Minor Comments:

1) Line 63 and in the references Zhang et al should be 2021 not 0009;
This has been corrected.

2) Line 168, I would specify “for eliciting heading responses in RSC if vision and proprioception are spared”.
This has been corrected.

3) About half of the ADN responses in figure 4D do not seem clearly unimodal. It is possible this variability originates from the head-fixation, however a histology confirmation of the injection target would provide more confidence.
Unfortunately the mice that were used for these experiments have been euthanized so we cannot confirm via histology the site of injection. However, no nearby areas to the ADN should project to the RSC, especially those that carry heading information, so the possibility that the bimodality is due to other subcortical projections is unlikely.

4) The authors use the word “landmark” but given the symmetric arrangement and the nature of the behavior, the authors should use “visual cue-driven” or “visual” cells.

We opted to call them “landmark” as opposed to “visual” cells because we wanted clarify that these cells likely are not receiving direct visual projections from the retina. The visual

information received by these cells is likely not a faithful representation of the external world -- instead representing a highly processed information stream that represents visual landmarks in the environment.

5) Line 221 it seems only 13% of the total cells are tuned – are these the “all RSC cells” shown in Fig 5A?

Yes that is correct.

6) Line 328 I would change heading to HD – the past literature mostly focused on the freely moving, vestibular- driven HD.

This has been corrected.

7) Line 445 typo, “independent”

This has been corrected.

8) Line 459 the sampling frequency is not mentioned.

This has been added.

9) Line 565 should be figure 5B.

This has been corrected.

10) Line 571 and later 577 the authors should cite the previous work they refer to.

These have been added.

11) Line 571 it is unclear what gls in the glsrvl acronym stands for

This has been corrected.

12) First part of equation 7 should have a minus sign.

This has been corrected.

13) Comparisons are provided for panels 5G and 5H, but not for 5I

Figure 5 has been updated.

14) Supplemental Fig3 example F has no detectable rates for the 180 degree, even though a small preference does appear in the polar plot on the right.

For clarity, we only show the first block of responses in cartesian coordinates, while the polar plot contains the response averaged over the imaging session. This point is clarified in the last sentence of the figure caption.

15) Is the pseudo-population decoding based on equal samples of the three populations or does it reflect the proportions in figure 5F (including the unclassified cells)?

The pseudo-population decoding is based on random sampling with replacement of all cells in the three classes, so the composition will roughly reflect the proportion of the response classes. This was clarified in the Methods section (lines 601-603).

REVIEWERS' COMMENTS

Reviewer #1 (Remarks to the Author):

I commend the author's substantial efforts to address my (and other reviewers) comments, including the additional clustering analyses pertaining to the axon imaging data and the additional experiments relating to passive cue rotations, CW and CCW rotations, and comparisons between controlled versus free floating conditions. It is my opinion that the alterations to the manuscript have added clarity. I believe that the scientific question is of great interest and that the experimental design is superb. That said, I believe there are remaining issues that should be addressed in rebuttal or a revised manuscript.

I continue to believe that the terms "coregistration" and "alignment" are nebulous and need more explicit definitions, preferably early in the manuscript.

Can we see the results from Figure 5I split into two plots for ADN and VIS axons and color coded according to cluster (as in Figure 5B) perhaps in a supplemental figure. As it stands, the current lack of quantification makes it unclear what proportion of alignment-like responses are present in AD or VIS terminals. As indicated in my previous review I believe this is a critical piece of the co-registration story presented.

Figure S7 is difficult to interpret, the legend states that landmark and alignment cells should have similar data yet alignment and heading look most similar.

As I suggested in my previous review, a reliability comparison for *each tuning peak* of the "alignment" versus "landmark" cells is important for supporting that the alignment cells are truly a separable population and not a noisier version of one another.

It is a bit confusing as to why the heading cell decoding in 6B is so inaccurate during light on when compared to landmark cells. Is the apparent difference statistically significant? If heading is accurately coded by these cells without vestibular information why is it so poor here?

Reviewer #2 (Remarks to the Author):

The authors have sufficiently addressed my concerns. I have no further comments.

Response to Reviewers

Reviewer #1 (Remarks to the Author):

I commend the author's substantial efforts to address my (and other reviewers) comments, including the additional clustering analyses pertaining to the axon imaging data and the additional experiments relating to passive cue rotations, CW and CCW rotations, and comparisons between controlled versus free floating conditions. It is my opinion that the alterations to the manuscript have added clarity. I believe that the scientific question is of great interest and that the experimental design is superb. That said, I believe there are remaining issues that should be addressed in rebuttal or a revised manuscript.

Thank you for your positive comments. We hope we have suitably addressed the comments below.

I continue to believe that the terms "coregistration" and "alignment" are nebulous and need more explicit definitions, preferably early in the manuscript.

We have added an explicit definition of these terms in the introduction (Lines 50 - 51)

Can we see the results from Figure 5I split into two plots for ADN and VIS axons and color coded according to cluster (as in Figure 5B) perhaps in a supplemental figure. As it stands, the current lack of quantification makes it unclear what proportion of alignment-like responses are present in AD or VIS terminals. As indicated in my previous review I believe this is a critical piece of the co-registration story presented.

This has been included as a new supplementary figure (new Figure S7).

Figure S7 is difficult to interpret, the legend states that landmark and alignment cells should have similar data yet alignment and heading look most similar.

We have removed the previous Figure S7 from the paper. In the original version of the analysis, the differences between each cell type was more pronounced. With the improved decoder and subsequent reduction of 180 degree decoding errors, we find that this figure no longer provides useful information.

As I suggested in my previous review, a reliability comparison for *each tuning peak* of the "alignment" versus "landmark" cells is important for supporting that the alignment cells are truly a separable population and not a noisier version of one another.

In order to test the reliability of each peak in landmark vs. alignment cells, we performed the following analysis: for each neuron in the visual cluster, we first took its average tuning curve in the light-on and light-off condition. We first align the tuning curves in a cross-validated method as performed elsewhere in the paper. Then, we calculate the reliability of each neuron's tuning

curve against the average visual tuning curve across all visual neurons, but specifically restricting the calculation to either a 10 bin window around the first or second peak (peak 1 and peak 2). This process is repeated in each condition, as well as for alignment neurons. The distributions of reliability as compared to the archetypal cluster for visual and alignment cells is similar in the light-on condition, but exhibit differences in the light-off condition, suggesting that they are separate populations (Reviewer Figure 1).

Reviewer Figure 1: Reliability of each peak to the average tuning curve. Histograms of reliability for alignment (left) vs visual (right) and light-on (top) vs light-off (bottom) tuning, split by each peak.

It is a bit confusing as to why the heading cell decoding in 6B is so inaccurate during light on when compared to landmark cells. Is the apparent difference statistically significant? If heading is accurately coded by these cells without vestibular information why is it so poor here?

The heading cell decoding performance is due to the relative unreliability of heading-cell responses without vestibular modulation. Because the decoder is dependent on the tuning curves, the decoder also performs more poorly as a result. We discuss these caveats in the discussion (Lines 316-320)

Reviewer #2 (Remarks to the Author):

The authors have sufficiently addressed my concerns. I have no further comments.

Thank you for your time and comments.